# Monte-Carlo Tree Search with Uncertainty Propagation via Optimal Transport

**Tuan Dam** [1]   **Pascal Stenger** [2]   **Lukas Schneider** [3]   **Joni Pajarinen** [4]   **Carlo D'Eramo** [2 5 6]
**Odalric-Ambrym Maillard** [7]

## Abstract

This paper introduces a novel backup strategy for Monte-Carlo Tree Search (MCTS) tailored for highly stochastic and partially observable Markov decision processes. We adopt a probabilistic approach, modeling both value and action-value nodes as Gaussian distributions, to introduce a novel backup operator that computes value nodes as the Wasserstein barycenter of their action-value children nodes; thus, propagating the uncertainty of the estimate across the tree to the root node. We study our novel backup operator when using a novel combination of $L^1$-Wasserstein barycenter with $\alpha$-divergence, by drawing a crucial connection to the generalized mean backup operator. We complement our probabilistic backup operator with two sampling strategies, based on optimistic selection and Thompson sampling, obtaining our Wasserstein MCTS algorithm. We provide theoretical guarantees of asymptotic convergence of $\mathcal{O}(n^{-1/2})$, with $n$ as the number of visited trajectories, to the optimal policy and an empirical evaluation on several stochastic and partially observable environments, where our approach outperforms well-known related baselines.

## 1. Introduction

Monte-Carlo Tree Search (MCTS) has become a crucial algorithmic paradigm for tackling challenging planning and Reinforcement Learning (RL) problems, particularly after its widespread success in deterministic games like Go and Chess (Silver et al., 2016a; 2017b). However, moving beyond these deterministic settings toward highly stochastic or partially observable Markov Decision Processes (MDPs/POMDPs) reveals major difficulties. In these cases, two key obstacles arise: *Uncertainty in Value Estimates:* In problems with substantial randomness or limited observability, naive value backups may lead to erroneous or unstable estimates, which propagate through the tree and degrade overall performance. *Exploration-Exploitation Balancing:* Traditional UCT-based exploration bonuses (Kocsis et al., 2006) can falter under high variance transitions, often causing either over- or under-exploration. Recent works (Tesauro et al., 2012; Bai et al., 2013; 2014) have suggested Bayesian or distributional methods for MCTS to better quantify uncertainty. Meanwhile, Metelli et al. (2019) leveraged $L^2$-Wasserstein barycenters to propagate distributional information in temporal-difference learning. Yet, several open questions remain on how to unify *distribution-based backups* and *flexible exploration strategies* within a single MCTS framework that provably handles high stochasticity and partial observability.

**Our Approach.** In this paper, we propose a new MCTS algorithm, *Wasserstein MCTS*, that models each node's value as a Gaussian distribution and propagates *both* mean and variance estimates throughout the tree. Crucially, we introduce a novel backup operator that computes *value nodes* as $L^1$-Wasserstein barycenters of their *action-value children*, using an $\alpha$-divergence as the distance measure. This yields:

- *Distributional Value Backups:* By tracking distributions (rather than point estimates), our method captures the inherent uncertainty of each node's value, especially valuable in stochastic or partially observable domains.

- *Generalized Mean Operator:* The $\alpha$-divergence ties naturally to the power-mean backup (Dam et al., 2019; 2024a), letting us interpolate between average-like and max-like updates to mitigate the overestimation often seen in RL (Hasselt, 2010).

We complement these distributional backups with two exploration mechanisms—an optimistic UCT bonus, and a Thompson sampling approach that selects actions by sam-

---

[1]Hanoi University of Science and Technology, Hanoi, Vietnam [2]Department of Computer Science, Technical University of Darmstadt, Germany [3]ETHZ - ETH Zurich, Switzerland [4]Department of Electrical Engineering and Automation, Aalto University, Finland [5]Center for Artificial Intelligence and Data Science, University of Würzburg, Germany [6]Hessian Center for Artificial Intelligence (Hessian.ai), Germany [7]Univ. Lille, Inria, CNRS, Centrale Lille, UMR 9189-CRIStAL, F-59000 Lille, France. Correspondence to: Tuan Dam <tuandq@soict.hust.edu.vn>.

*Proceedings of the 42$^{nd}$ International Conference on Machine Learning*, Vancouver, Canada. PMLR 267, 2025. Copyright 2025 by the author(s).

pling from the node's Gaussian posterior.

**Our Key Contributions.** *1. Uncertainty Propagation via $L^1$-Wasserstein Barycenters.* We provide a principled way to back up distributions in an MCTS, unifying $L^1$-Wasserstein geometry and $\alpha$-divergences to handle high variance and partial observability. *2. Connection to Generalized Mean Backup.* Our backup operator yields a power-mean update for node values, enabling a controllable continuum between overly optimistic (max-like) and risk-averse (average-like) estimates. *3. Polynomial Convergence Analysis.* We prove that *Wasserstein MCTS* with Thompson sampling converges to the optimal policy at a rate $\mathcal{O}(n^{-1/2})$, matching known lower bounds. This is in contrast to prior distributional MCTS methods that lacked explicit convergence guarantees. *4. Extensive Empirical Validation.* On a suite of highly stochastic MDPs (e.g. *River-Swim, Taxi*) and partially observable tasks (*Pocman, Rocksample*), our approach outperforms established baselines, including `UCT`, `Power-UCT`, and Bayesian MCTS variants.

Overall, *Wasserstein MCTS* offers a flexible and theoretically grounded framework for handling uncertainty within MCTS. By combining Gaussian node models, $L^1$-Wasserstein barycenters, and $\alpha$-divergences, it effectively balances exploration and exploitation in domains where noise or partial observability make traditional MCTS methods brittle.

## 2. Related Work

Metelli et al. (2019) use $L^2$-Wasserstein barycenters to propagate uncertainty in temporal-difference learning. In MCTS, Bayesian methods handle uncertainty by treating values as Gaussian distributions (Tesauro et al., 2012) or Dirichlet-NormalGamma posteriors (Bai et al., 2013; 2014). Unlike these, we propagate uncertainty *throughout* the tree via $L^1$-Wasserstein barycenters and $\alpha$-divergences, linking to generalized-mean backups (Dam et al., 2019) and maintaining both mean and variance estimates. This distributional perspective is effective in highly stochastic or partially observable tasks. In multi-armed bandits, optimism (Auer et al., 2002a) and Thompson sampling (Thompson, 1933) are standard; we combine these with our uncertainty propagation scheme to guide action selection in MCTS.

## 3. Background

### 3.1. Markov Decision Process

We consider an agent in an infinite-horizon discounted Markov decision process (MDP) $\mathcal{M} = \langle \mathcal{S}, \mathcal{A}, \mathcal{R}, \mathcal{P}, \gamma \rangle$, where $\mathcal{S}$ is the state space, $\mathcal{A}$ is the finite action space, $\mathcal{R} : \mathcal{S} \times \mathcal{A} \times \mathcal{S} \rightarrow \mathbb{R}$ is the reward function,

$\mathcal{P} : \mathcal{S} \times \mathcal{A} \rightarrow \mathcal{S}$ is the transition kernel, and $\gamma \in [0, 1)$ is the discount factor. A policy $\pi \in \Pi : \mathcal{S} \rightarrow \mathcal{A}$ defines the action selection probabilities based on states. The action-value function $Q^\pi$ is given by $Q^\pi(s, a) \triangleq \mathbb{E}\left[\sum_{k=0}^{\infty} \gamma^k r_{i+k+1} \mid s_i = s, a_i = a, \pi\right]$, representing the expected cumulative discounted reward for executing action $a$ in state $s$ and following policy $\pi$. The objective is to find the optimal policy that maximizes $Q^\pi$, satisfying the Bellman equation (Bellman, 1954): $Q^*(s, a) \triangleq \int_{\mathcal{S}} \mathcal{P}(s'|s, a)\left[\mathcal{R}(s, a, s') + \gamma \max_{a'} Q^*(s', a')\right] ds', \quad \forall s \in \mathcal{S}, a \in \mathcal{A}$. From the optimal action-value function, we derive the optimal value function as $V^*(s) \triangleq \max_{a \in \mathcal{A}} Q^*(s, a), \quad \forall s \in \mathcal{S}$.

### 3.2. Monte-Carlo Tree Search

Monte-Carlo Tree Search (MCTS) combines Monte-Carlo sampling, tree search, and exploration strategies from multi-armed bandits (Auer et al., 2002b) to solve MDPs. It builds a search tree where states are nodes and actions are edges. MCTS involves four key steps: Selection: Navigate from the root to a leaf node using a *tree-policy*. Expansion: Expand the reached node based on the tree policy. Simulation: Perform a rollout (Monte-Carlo simulation) from the child node to estimate its value, or use a pretrained neural network (Silver et al., 2016a) for this estimation. Backup: Update the action-values $Q(\cdot)$ along the visited trajectory using the collected rewards.

## 4. Formalization

**Problem Setup** Monte Carlo Tree Search (MCTS) is an algorithm for exploring and evaluating trajectories in an MDP. Starting from an initial state $s_0$, MCTS incrementally builds a planning tree by simulating trajectories. Each trajectory either reaches a leaf node or terminates when a predetermined maximum depth $H$ is reached. At the end of each trajectory, a *playout policy* (which may be deterministic or stochastic) is executed from the final node reached, allowing the algorithm to evaluate the associated state. After running for $t$ trajectories, the MCTS algorithm provides the following outputs:

- $\overline{a}_t$: estimate of the optimal action to take in state $s_0$,
- $\overline{V}_t(s_0)$: estimate of the optimal value function at $s_0$.

**Evaluating MCTS Performance** The performance of the MCTS algorithm is assessed based on its *convergence rate*, $r(t)$, which quantifies how quickly the algorithm approaches the optimal policy. Specifically, the following bounds hold:

$$\mathbb{E}\left[V^\star(s_0) - Q^\star(s_0, \overline{a}_t)\right] \leqslant r(t),$$
$$\left|\mathbb{E}\left[V^\star(s_0) - \overline{V}_t(s_0)\right]\right| \leqslant r(t),$$

where $V^\star(s_0)$ and $Q^\star(s_0, a)$ are the true optimal value and action-value functions at state $s_0$, respectively.

**Recursive Value Estimation** To analyze the MCTS algorithm, we consider a planning horizon $H$ and a playout policy $\pi_0$ with an associated value function $V_0$. For each node $s_h$ at depth $h$ (i.e., the state reached after $h$ steps from $s_0$), we recursively define the value function $\widetilde{V}(s_h)$ as follows. At the leaf nodes ($h = H$), the value function is simply the playout policy's value:

$$\widetilde{V}(s_H) = V_0(s_H).$$

For all other depths $h \leqslant H - 1$, we compute the action-value function $\widetilde{Q}(s_h, a)$ and value function $\widetilde{V}(s_h)$ as:

$$\widetilde{Q}(s_h, a) = r(s_h, a) + \gamma \sum_{s_{h+1} \in \mathcal{A}_s} \mathcal{P}(s_{h+1} \mid s_h, a)\widetilde{V}(s_{h+1}),$$

$$\widetilde{V}(s_h) = \max_a \widetilde{Q}(s_h, a),$$

where $r(s_h, a)$ is the mean immediate reward obtained by taking action $a$ in state $s_h$, $\mathcal{P}(s_{h+1} \mid s_h, a)$ is the probability of transitioning to state $s_{h+1}$ from $s_h$ given action $a$ and $\gamma$ is the discount factor.

**Bounding the Error** The recursive structure of the value estimates gives rise to a bound on the error between the true optimal action-value function $Q^\star(s_0, a)$ and the MCTS estimate $\widetilde{Q}(s_0, a)$. Specifically, we have:

$$\left| Q^\star(s_0, a) - \widetilde{Q}(s_0, a) \right| \leqslant \gamma^H \|V^\star - V_0\|_\infty,$$

where the supremum norm $\|V^\star - V_0\|_\infty$ can be restricted to states reachable within $H$ steps from $s_0$.

**Goal of MCTS** The ultimate aim of the MCTS algorithm is to minimize the convergence rate $r(t)$ by constructing accurate estimates of $\widetilde{Q}(s_0, a)$ and $\widetilde{V}(s_0)$, which in turn approach the true optimal functions $Q^\star(s_0, a)$ and $V^\star(s_0)$, and then identify the best action at the root node:

$$a_\star = \arg \max_a Q^\star(s_0, a).$$

# 5. Wasserstein Barycenter With $\alpha$-Divergence

We introduce the key notions behind our distribution-based backups: the *Wasserstein barycenter* and the *$\alpha$-divergence*. Unlike prior works that use $L^2$-based Wasserstein distances (Metelli et al., 2019), we adopt an $L^1$-Wasserstein distance combined with $\alpha$-divergences. This combination yields more robust value backups in stochastic and partially observable settings.

## 5.1. Wasserstein Barycenter

Let $(\mathcal{X}, d)$ be a Polish (complete, separable metric) space. For $q \geq 1$, define $\mathcal{P}_q(\mathcal{X})$ as the set of probability measures

on $\mathcal{X}$ whose $q$-th moment is finite. For two distributions $\mu, \nu \in \mathcal{P}_q(\mathcal{X})$, the *$L^q$-Wasserstein distance* is

$$W_q(\mu, \nu) = \left( \inf_{\rho \in \Gamma(\mu, \nu)} \mathbb{E}_{(X,Y) \sim \rho}\left[ d(X, Y)^q \right] \right)^{1/q},$$

where $\Gamma(\mu, \nu)$ is the set of joint couplings whose marginals match $\mu$ and $\nu$. Given $n$ distributions $\{\nu_i\}_{i=1}^n$ and weights $\{w_i\}$ summing to 1, the *$L^q$-Wasserstein barycenter* is

$$\bar{\nu} = \arg \min_\nu \sum_{i=1}^n w_i W_q(\nu, \nu_i)^q.$$

Our work focuses on $q = 1$.

## 5.2. $\alpha$-divergence and the $L^1$ Wasserstein Barycenter

In many distribution-based backup schemes, the *Wasserstein distance* is a natural choice to quantify how "far apart" two distributions are. A commonly used approach (Metelli et al., 2019) is to employ the $L^2$-Wasserstein metric. In contrast, we consider an $L^1$-Wasserstein formulation coupled with an $\alpha$-divergence for two main reasons:

- *Robustness & Aggregation Control.* An $L^1$-based metric can be more robust to outliers and large deviations than $L^2$. Furthermore, combining it with the $\alpha$-divergence allows a continuous interpolation between averaging and max-like backups (through the $\alpha$ parameter).

- *Connection to Power-Mean Updates.* Modeling nodes as Gaussians (or particle distributions) and relying on $L^1$-Wasserstein with an $\alpha$-divergence yields closed-form updates that coincide with the power-mean operator. This unifies average and maximum backups in a single formula and lets us propagate both means and variances (uncertainty) through the tree.

**$f$-divergences and the $\alpha$-divergence.** An $f$-divergence (Csiszár, 1964) between two points $X$ and $Y$ over a Manifold $\mathcal{M}$ defined as

$$D_{f_\alpha}(X\|Y) = \sum_i \xi_Y^{(i)} f_\alpha\left( \frac{\xi_X^{(i)}}{\xi_Y^{(i)}} \right), \quad f_\alpha(x) = \frac{x^\alpha - 1 - \alpha(x-1)}{\alpha(\alpha-1)},$$

where varying $\alpha$ controls how aggressively or conservatively we measure the "distance" between $X$ and $Y$.

**Constructing the $L^1$-Wasserstein Barycenter.** In our approach, the $L^1$-Wasserstein distance between $\nu$ and $\nu_i$ is defined via

$$W_1(\nu, \nu_i) = \inf_{\rho \in \Gamma(\nu, \nu_i)} \mathbb{E}_{(X,Y) \sim \rho}\left[ D_{f_\alpha}(X, Y) \right]. \quad (1)$$

The $L^1$-Wasserstein barycenter then solves

$$\bar{\nu} = \arg \inf_\nu \left\{ \sum_{i=1}^n w_i W_1(\nu, \nu_i) \right\},$$

i.e., we seek the single distribution $\bar{\nu}$ that jointly minimizes its $L^1$-Wasserstein distance (defined via the $\alpha$-divergence) to all the $\nu_i$.

**Why $L^1$ instead of $L^2$.** Using the $L^1$ distance in equation 1 naturally leads to a backup rule resembling the *power mean* operator Proposition 1. This power-mean update is more robust to high-variance samples and connects smoothly to both the average backup (when $\alpha \to 0$ or $p = 1$) and the max backup (as $\alpha \to \infty$ or $p \to \infty$). Hence, $L^1$-Wasserstein with $\alpha$-divergences offers a principled way to blend distributions in highly stochastic environments while controlling the balance between underestimation and overestimation in the final backup.

**Why Use an $\alpha$-Divergence Instead of $L^2$?** Although $\alpha$-divergences are not strict metrics (they can be asymmetric and need not satisfy the triangle inequality), their use within an $L^1$-Wasserstein framework provides distinct benefits for MCTS under stochastic or partially observable conditions:

- *Greater Flexibility via Generalized Means.* When combined with the $L^1$-Wasserstein distance, an $\alpha$-divergence naturally yields a *power-mean* style backup operator (Dam et al., 2019). By adjusting the parameter $\alpha$, one smoothly interpolates between average-like and max-like backups, allowing precise control over how conservative or aggressive the updates should be. This stands in contrast to $L^2$-based distances, which only yield fixed (e.g. purely quadratic) aggregation behavior.

- *Robustness to Stochastic Variations.* Because $\alpha$-divergences can emphasize or de-emphasize portions of the distribution differently depending on $\alpha$, they help mitigate overestimation or underestimation in highly stochastic settings. Empirical studies in distributional RL (Metelli et al., 2019) suggest that more adaptive divergence measures can significantly improve stability and performance when the underlying dynamics involve heavy noise.

- *No Need for Symmetry in Backups.* MCTS requires a *cost functional* to aggregate posterior distributions across children nodes, rather than a strict metric. Hence, the lack of symmetry or the triangle inequality does not undermine its validity here. An $f$-divergence—including $\alpha$-divergences—is sufficient to drive consistent updates of belief distributions in the tree.

- *Unified Framework for Various Divergences.* The $\alpha$-divergence family subsumes and generalizes many standard divergences (e.g. KL, reverse KL). This single-parameter approach enables users to easily switch or fine-tune the update behavior for different problem characteristics, rather than designing separate algorithms for each divergence.

- *Direct Theoretical Connections.* Under mild assumptions, $L^1$-Wasserstein geometry paired with $\alpha$-divergences admits closed-form or near-closed-form power-mean formulas (Dam et al., 2019). This not only streamlines theoretical analysis but also simplifies implementation by allowing straightforward computation of mean and variance updates at each node.

In practice, these properties make $\alpha$-divergences well-suited for uncertainty propagation within MCTS: despite not being a metric, their adaptability and connection to generalized means allow them to effectively handle complex, high-variance environments.

### 5.3. V-posterior

It is natural to define a value node as the V-posterior computed with $L^1$-Wasserstein barycenters of the children nodes Q-posteriors, following a procedure inspired by Metelli et al. 2019 (Metelli et al., 2019) and tailored to MCTS.

**Definition 1** (V-posterior)**.** *Given a policy $\bar{\pi}$ and a state $s \in \mathcal{S}$, we define the V-posterior $\mathcal{V}(s)$ induced by Q-posteriors $\mathcal{Q}(s, a)$ with $a \in \mathcal{A}$ as the $L^1$-Wassertein barycenter of the $\mathcal{Q}(s, a)$:*

$$\mathcal{V}(s) \in \arg\inf_{\mathcal{V}} \left\{ \mathbb{E}_{a \sim \bar{\pi}(.|s)} \big[ W_1(\mathcal{V}, \mathcal{Q}(s, a)) \big] \right\}.$$

In this work, we model each node in the tree as a Gaussian distribution. We define $p = 1 - \alpha$ and derive the following.

**Proposition 1.** *Consider the V-posterior value function $\mathcal{V}(s)$ as a Gaussian: $\mathcal{N}(\overline{m}(s), \overline{\sigma}^2(s))$. Define each $\mathcal{Q}(s, a)$ as the action-value function child node of $\mathcal{V}(s)$. Each $\mathcal{Q}(s, a)$ is assumed as a Gaussian distributions $\mathcal{Q}(s, a) : \mathcal{N}(m(s, a), \sigma(s, a)^2)$. If the value function $\mathcal{V}(s)$ is defined as the Wasserstein barycenter of the action-value function $\mathcal{Q}(s, a)$, given the policy $\bar{\pi}$, we have*

$$\overline{m}(s) = (\mathbb{E}_{a \sim \bar{\pi}}[m(s, a)^p])^{\frac{1}{p}}$$
$$\overline{\delta}(s) = (\mathbb{E}_{a \sim \bar{\pi}}[\delta(s, a)^p])^{\frac{1}{p}}.$$

Proposition 1 shows the closed form solutions of the mean and standard deviation of the Gaussian value function $\mathcal{V}(s)$ considering it as the $L^1$-Wasserstein barycenter Q-posteriors. In detail, the mean of $\mathcal{V}(s)$ are the power mean of all mean values of all the $\mathcal{Q}(s, a)$ function, considering the finite set of actions. When $p = 1$, we derive the expected form solutions.

We point out that our approach is not restricted to the Gaussian distribution model. We get the following result by considering each tree node as a particle model.

**Proposition 2.** *Consider the V-posterior value function $\mathcal{V}(s)$ as an equally weighted Particle model: $\overline{x_i}(s) : i \in [1, M]$. $M$ is an integer and $M \geqslant 1$. Assume each action-value function $\mathcal{Q}(s, a)$ has $M$ particles $x_i(s, a), i \in [1, M]$. If the value function $\mathcal{V}(s)$ is defined as the Wasserstein barycenter of the action-value function $\mathcal{Q}(s, a)$, given the policy $\bar{\pi}$, each particle $\overline{x_i}(s), i \in [1, M]$ can be estimated as*

$$\overline{x_i}(s) = (\mathbb{E}_{a \sim \bar{\pi}}[x_i(s, a)^p])^{1/p},$$

Proposition 2 shows that each particle of the V-posterior value function $\mathcal{V}(s)$ can be derived as the power mean of the respective particles of all the $\mathcal{Q}(s, a)$ function. If $p = 1$, we again get the closed-form solutions as the expectation of the respective particles of all the $\mathcal{Q}(s, a)$ functions. The results in Proposition 1, and Proposition 2 can be considered as the generalized result of Proposition A.3 in Metelli et al. (2019). In the next section, we present our Wasserstein Monte-Carlo tree search (`W-MCTS`) algorithm, assuming each tree node is a Gaussian distribution.

# 6. Wasserstein Monte-Carlo Tree Search

We introduce our Wasserstein Monte-Carlo Tree Search (`W-MCTS`), where V-posteriors are modeled as Wasserstein barycenters of action-value distributions. With Gaussian distributions at each node, we define backup operators for mean and variance. Additionally, we propose two action selection strategies: optimistic selection and Thompson sampling.

## 6.1. Backup Operator

We model each $V$-node and $Q$-node as a Gaussian with mean and standard deviation:

$$V_{\mathrm{m}}(s), \ V_{\mathrm{std}}(s) \quad \text{and} \quad Q_{\mathrm{m}}(s, a), \ Q_{\mathrm{std}}(s, a).$$

We denote $\overline{V}_{\mathrm{m}}(s, N(s))$ as the *empirical mean estimate* of the $V$-node at state $s$ after $N(s)$ total visits, and $\overline{Q}_{\mathrm{m}}(s, a, n(s, a))$ as the *empirical mean estimate* of the $Q$-node at $(s, a)$ after $n(s, a)$ visits. Likewise, $\overline{V}_{\mathrm{std}}(s, N(s))$ and $\overline{Q}_{\mathrm{std}}(s, a, n(s, a))$ are their corresponding *empirical standard deviation estimates*.

$V$-**nodes.** From Proposition 1, the mean and the standard deviation of a $V$-node is a power-mean aggregation of its $Q$-children:

$$\overline{V}_{\mathrm{m}}(s, N(s)) \ \leftarrow \ \left( \sum_a \frac{n(s, a)}{N(s)} \left[ \overline{Q}_{\mathrm{m}}(s, a, n(s, a)) \right]^p \right)^{1/p},$$

$$\overline{V}_{\mathrm{std}}(s, N(s)) \ \leftarrow \ \left( \sum_a \frac{n(s, a)}{N(s)} \left[ \overline{Q}_{\mathrm{std}}(s, a, n(s, a)) \right]^p \right)^{1/p},$$

where $n(s, a)$ is the visit count of action $a$ at state $s$, and $N(s) = \sum_a n(s, a)$. For $p = 1$, this reduces to the standard average, whereas $p > 1$ induces a more "max-like" backup (Dam et al., 2019).

$Q$-**nodes.** Under the Bellman-style backup for each $Q$-node,

$$Q_{\mathrm{m}}(s, a) = \mathbb{E}[r(s, a)] + \gamma \, \mathbb{E}[V_{\mathrm{m}}(s')], Q_{\mathrm{std}}(s, a) = \gamma \, V_{\mathrm{std}}(s'),$$

we replace expectations by empirical sums and visitation counts:

$$\overline{Q}_{\mathrm{m}}(s, a, n(s, a)) \ \leftarrow \ \frac{\sum r(s, a) + \gamma \sum_{s'} N(s') \, \overline{V}_{\mathrm{m}}(s', N(s'))}{n(s, a)},$$

$$\overline{Q}_{\mathrm{std}}(s, a, n(s, a)) \ \leftarrow \ \frac{\gamma \sum_{s'} N(s') \, \overline{V}_{\mathrm{std}}(s')}{n(s, a)}.$$

Here, the sums range over transitions and children states $s'$, weighted by their visit counts $N(s')$. As $n(s, a)$ grows large, both the variance and mean estimators stabilize, eventually converging to deterministic values.

## 6.2. Action Selection

Monte Carlo Tree Search can adopt a variety of exploration strategies based on the original `UCT` framework (Kocsis et al., 2006). In practice, multiple refinements exist, such as the variants used in AlphaGo (Silver et al., 2016b), AlphaZero (Silver et al., 2017c;a), MuZero (Schrittwieser et al., 2020), Stochastic MuZero (Antonoglou et al., 2021), and Stochastic-Power-UCT (Dam et al., 2024b). Although different choices of the exploration constant or bonus lead to different performance characteristics, we retain the standard, state-of-the-art designs described below. In our theoretical analysis, however, we focus specifically on Thompson sampling, since the `UCT`-like optimistic selection can be viewed as a special case of the well-studied Power-UCT algorithm (Dam et al., 2019; 2024b).

**Optimistic Selection.** A classic `UCT`-style selection picks actions using upper confidence bounds on $Q$-values,

$$a \ = \ \operatorname*{argmax}_{a_i} \left[ m(s, a_i) \ + \ C \sqrt{\frac{\log N(s)}{n(s, a_i)}} \right],$$

where $m(s, a_i)$ is the empirical mean, $n(s, a_i)$ is the visit count of action $a_i$, and $N(s)$ is the total visit count at state $s$. Replacing the $\frac{1}{\sqrt{n(s, a_i)}}$ term by the empirical standard deviation $\sigma(s, a_i)$ yields an *optimistic* variant of Wasserstein MCTS (`W-MCTS-OS`):

$$a \ = \ \operatorname*{argmax}_{a_i} \left[ m(s, a_i) \ + \ C \, \sigma(s, a_i) \sqrt{\log N(s)} \right].$$

The factor $\sigma(s, a_i) \approx 1/\sqrt{n(s, a_i)}$ follows from a CLT-based argument.

**Thompson Sampling.** In contrast, Thompson sampling stochastically samples an action from the $Q$-posterior:

$$a = \operatorname*{argmax}_{a_i} \big\{ \theta_i \sim \mathcal{N}\big( m(s, a_i), \sigma^2(s, a_i) \big) \big\}.$$

We refer to this Thompson variant as *Wasserstein MCTS-TS* (`W-MCTS-TS`). In Section 7, we analyze its convergence properties under non-stationary multi-armed bandits and then leverage these results to establish convergence in the planning tree.

# 7. Theoretical Analysis

## 7.1. Analysis Setup

We define the setting for our theoretical analysis using a class of non-stationary Multi-Armed Bandit (MAB) problems at each state $s$ in the MCTS tree. Consider $K$ arms (actions), each with a mean reward $\mu_k$, for $k \in [K]$. At time step $t$, pulling arm $k$ yields a random reward $X_{k,t}$, bounded within $[0, R]$. The average reward for arm $k$ after $n$ trials is:

$$\overline{X}_{k,n} = \frac{1}{n} \sum_{t=1}^{n} X_{k,t}, \quad \text{with} \quad \mu_{k,n} = \mathbb{E}[\overline{X}_{k,n}]$$

Let $\star$ represent quantities related to the optimal arm, and denote $T_k(n)$ as the number of times arm $k$ has been played by step $n$. We assume the following *concentration* condition holds:

**Assumption 1.** *We assume that the reward sequence, $\{X_{k,t} : t \geqslant 1\}$, is a non-stationary process satisfying the assumption: for all $1 > \varepsilon > 0, \exists c > 0$ that*

$$\mathbf{Pr}\left( |\overline{X}_{k,n} - \mu_k| > \varepsilon \right) \leqslant c n^{-1} \varepsilon^{-2}, k \in [K]. \quad (2)$$

## 7.2. Main Results

We show the polynomial convergence of the expected estimated mean value function at the root node in Theorem 1.

### 7.2.1. CONVERGENCE OF `W-MCTS`

We start with an important result as shown below

**Proposition 3.** *Applying `W-MCTS` to an MCTS tree of depth $(H)$, at any depth h of the tree, we have*

*(i) At any depth $h$, $\exists$ constant $C_0 > 0$ that for any $0 < \varepsilon < 0, n \geqslant 1$, we can derive*

$$\mathbf{Pr}\bigg( \Big| \overline{V}_m(s_h, a_k, n) - \widetilde{V}(s_h, a_k) \Big| \geqslant \varepsilon \bigg) \leqslant C_0 n^{-1} \varepsilon^{-2}.$$

*(ii) At any depth $h$, $\exists$ constant $C_0 > 0$ that for any $0 < \varepsilon < 0, n \geqslant 1$, we can derive*

$$\mathbf{Pr}\bigg( \Big| \overline{Q}_m(s_h, a_k, n) - \widetilde{Q}(s_h, a_k) \Big| \geqslant \varepsilon \bigg) \leqslant C_0 n^{-1} \varepsilon^{-2}.$$

Proof Sketch

**MCTS as a Hierarchical Bandit Structure.** The Monte Carlo Tree Search (MCTS) algorithm can be viewed as a hierarchy of multi-armed bandits (MABs), where each node in the search tree represents an independent bandit problem. In this framework, the reward for each node, or current bandit, is influenced by the performance of the bandit algorithms applied to its child nodes. Since the `W-MCTS` policy adapts dynamically to balance exploitation and exploration, the rewards at each node are inherently *non-stationary*. The proof of Theorem 1 unfolds through three essential steps:

**1. Analyzing Non-stationary Bandits** The initial step focuses on the analysis of a non-stationary multi-armed bandit, which reflects the behavior of MABs at each MCTS node. We establish that if the rewards of these non-stationary bandits meet specific *concentration* properties, the regret induced by the `W-MCTS` algorithm will exhibit corresponding concentration guarantees. This outcome is formally stated in Theorem 2.

**2. Induction Argument** Next, we utilize an inductive argument to transfer the convergence and concentration properties from the lower tree levels to the root node. As the rewards from one level inform those of the next, the findings from Step 1 can be recursively applied. We begin at depth $H-1$ and move upward, demonstrating inductively that the bandit rewards at each level $H$ of the MCTS satisfy the criteria required by Theorem 2. This process propagates the desired properties up to the root node, completing the induction.

**3. Error Analysis from the Oracle** The final step examines the error introduced by the leaf node estimator, represented by the value function oracle $V_0$. With this oracle, the depth-$H$ MCTS can be interpreted as performing $H$ steps of value iteration, starting from $V_0$ at the leaf nodes (as mentioned in (Dam et al., 2024b)). Importantly, the oracle's error decreases geometrically at a rate of $\gamma$ due to the contraction mapping property of value iteration, leading to diminishing error as we ascend from the leaf nodes to the root. The complete proof for Proposition 3 can be found in the supplemental material. Finally, we get the main result.

**Theorem 1.** *We have at the root node $s_0$,*

$$\Big| \mathbb{E}[\overline{V}_m(s_0, n)] - \widetilde{V}(s_0) \Big| \leqslant \mathcal{O}(n^{-1/2}).$$

Our proposed method, `W-MCTS`, achieves a polynomial convergence rate of $\mathcal{O}(n^{-1/2})$, matching the results of Dam et al. (2024b). In contrast, Xiao et al. (2019) introduced `MENTS`, followed by `RENTS` and `TENTS` from Dam et al. (2021), which leverage exponential convergence to a

regularized value function through maximum entropy regularization. However, these methods face bias due to errors in the regularized value function, potentially leading to incorrect action selection. Conversely, Painter et al. (2024) employ a similar action selection strategy with a maximum backup operator for value estimation, resulting in exponential reductions in simple regret. However, their method's effectiveness heavily relies on the temperature parameter in Boltzmann exploration, limiting its practical use.

### 7.2.2. WASSERSTEIN NON-STATIONARY MULTI-ARMED BANDIT

A crucial part of the proof for Theorem 1 is to derive the following result for the `W-MCTS` in bandit setting. Under the Assumption 1, we consider applying Thompson Sampling strategy as the action selection method for the non-stationary multi-armed bandit (MAB) problems describes above. At each time step $n$, an action is selected as

$$a = \underset{a_i, i \in \{1...K\}}{\mathrm{argmax}} \{\theta_i \sim \mathcal{N}(\overline{X}_{k,n}, V_k/T_k(n))\}. \quad (3)$$

Let's define $\overline{X}_n(p) = \left( \sum_{a=1}^{K} \left( \frac{T_a(n)}{n} \right) \overline{X}_{a,T_a(n)}^p \right)^{1/p}$ as the power mean value backup at the root node, $T_a(n) = \sum_{t=1}^{n-1} \mathbb{1}(a_t = a)$ is the number of selections of $a$ prior to round $n$. We show theoretical results of our method as follows. Under the Assumption 1, we establish the concentration properties of the power mean backup operator $\overline{X}_n(p)$ towards the mean value of the optimal arm $\mu_* = \max_a \{\mu_a\}, a \in [K]$, as shown in Theorem 2.

**Theorem 2.** *Consider a non-stationary bandit problem described as in 7.1 with action selection as Equation (3). Then,*

$$\mathbf{Pr}(|\overline{X}_n(p) - \mu_\star| \geqslant \varepsilon) \leqslant Cn^{-1}\varepsilon^{-2}.$$

Theorem 2 states the concentration properties of the power mean estimation by `W-MCTS` for a non-stationary continuous-armed bandit problem, and play an important role for the induction proof of Proposition 3 leading to the main result presented at Theorem 1.

## 8. Experiments

### 8.1. Fully Observable, Highly Stochastic Tasks

We compare `W-MCTS` to UCT (Kocsis et al., 2006), `Power-UCT` (Dam et al., 2019), and DNG (Bai et al., 2013) in five benchmark environments: *FrozenLake*, *NChain*, *RiverSwim*, *SixArms*, and *Taxi*. These tasks all feature significant stochasticity or long-horizon exploration challenges.

**FrozenLake.** A $4 \times 4$ grid with slippery transitions, implemented in OpenAI Gym (Brockman et al., 2016). The

agent aims to reach a goal in the bottom-right corner. Due to frequent slips, each move has high uncertainty. Figure 1 shows that `W-MCTS-TS` (Thompson sampling) outperforms DNG, UCT, `Power-UCT`, and `W-MCTS` (optimistic selection), with `W-MCTS` at $p = 1$ performing comparably to `W-MCTS-TS`.

**NChain.** An agent can move forward or backward along a chain of length 5. Actions may reverse with 20% probability, making consistent forward progress difficult. In Figure 1, both `W-MCTS-TS` and `W-MCTS-OS` exceed UCT and `Power-UCT` in convergence speed and final returns.

**RiverSwim.** Similar to *NChain* but more complex transitions: sometimes the agent remains in the same state or only partially moves. This rewards long-term planning to reach high-value states. As in Figure 1, `W-MCTS-OS` converges fastest and attains the best performance, while `Power-UCT` eventually reaches similar returns more slowly.

**SixArms.** A 7-state chain with 6 possible arms (actions) leading to different rewards that scale inversely with their success probabilities. This environment demands high exploration. Figure 1 shows that `W-MCTS` is the only method consistently securing strong returns.

**Taxi.** A $7 \times 6$ grid where the agent must pick up three passengers, then reach a goal region. Slips occur 10% of the time, adding further uncertainty. Only `W-MCTS-TS` manages to collect all passengers reliably, outperforming `Power-UCT` and `W-MCTS` with optimistic selection.

### 8.2. Partially Observable, Highly Stochastic Tasks

We also test `W-MCTS` against `POMCP(UCT)`, D2NG, and DESPOT in classic POMDP benchmarks: *rocksample*, *pocman*, *Tag*, and *LaserTag*. Code for `POMCP(UCT)` (Silver & Veness, 2010b), D2NG (Bai et al., 2014), and DESPOT (Somani et al., 2013) is used as released by the original authors.

**Rocksample.** A robot on an $n \times n$ grid can sample or ignore $k$ rocks, then exit. We test three variants: (11,11), (15,15), and (15,35). Figure 2 shows that `W-MCTS-TS` consistently outperforms UCT and D2NG.

**Pocman.** A partially observed maze (Silver & Veness, 2010a) where the agent must collect pellets while avoiding ghosts. Table 1 indicates that `W-MCTS-TS` with $p = 100$ outperforms UCT and D2NG across most rollout-budget settings, and `W-MCTS-OS` also matches or surpasses these baselines in some configurations.

**Comparison with DESPOT.** We additionally compare `W-MCTS` to DESPOT across *Tag*, *LaserTag*, *rocksample* $(15 \times 15)$, and *Pocman*. Table 2 shows that `W-MCTS-OS` and `W-MCTS-TS` achieve higher returns than AB-DESPOT and AR-DESPOT in rocksample. Similarly, `W-MCTS-TS` surpasses DESPOT in *Pocman*, *Tag*, and *LaserTag*, while `W-MCTS-OS` outperforms AB-DESPOT in *Pocman*. **Role**

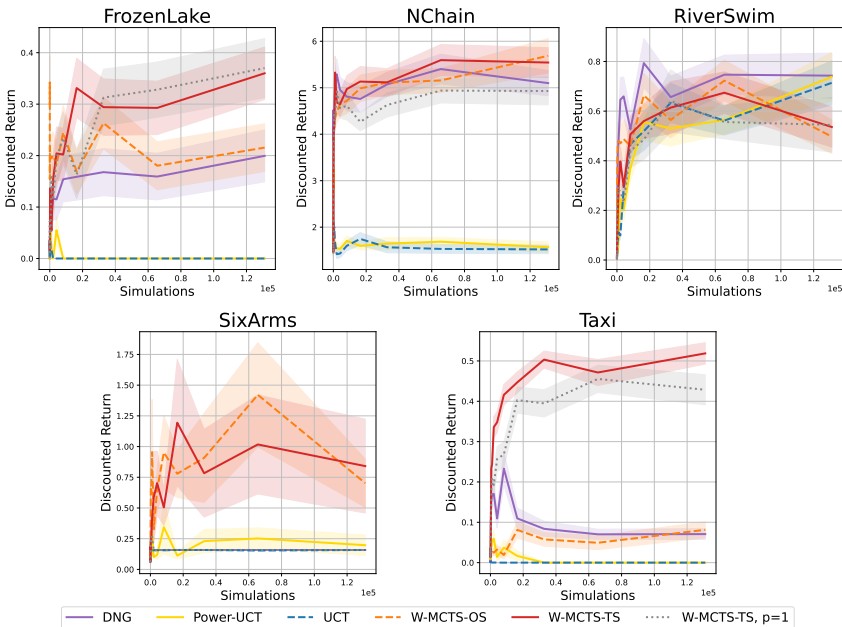

Figure 1: Performance of `W-MCTS` vs. `DNG`, `Power-UCT`, and `UCT` on five MDPs. Each curve shows the mean discounted return (averaged over 50 runs), with shaded regions indicating standard error.

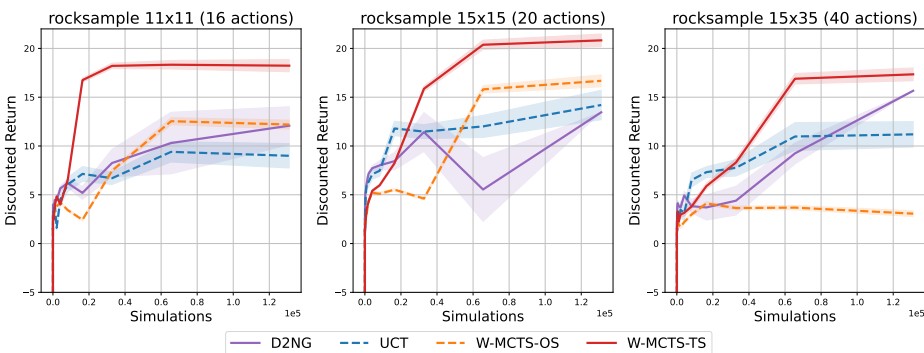

Figure 2: Performance of `W-MCTS` vs. `D2NG` in *rocksample*, averaged over 1000 runs (except `UCT`, 100 runs). Shaded areas denote standard error.

Table 1: Discounted total reward in *pocman*. Mean ± standard error are computed from 1000 random seeds.

|  | 1024 | 4096 | 32768 | 65536 |
|---|---|---|---|---|
| `W-MCTS-OS`, $p = 1$ | $50.9 \pm 0.6$ | $51.0 \pm 0.62$ | $52.2 \pm 0.79$ | $54.6 \pm 1.08$ |
| `W-MCTS-TS`, $p = 100$ | $67.38 \pm 0.53$ | $\mathbf{75.64 \pm 0.51}$ | $\mathbf{77.68 \pm 0.77}$ | $\mathbf{77.70 \pm 1.22}$ |
| `D2NG` | $\mathbf{71.55 \pm 0.57}$ | $75.39 \pm 1.47$ | $76.90 \pm 6.40$ | $72.2 \pm 0.0$ |
| `UCT` | $23.4 \pm 0.99$ | $23.6 \pm 1.09$ | $24.90 \pm 3.40$ | $28.5 \pm 3.8$ |

Table 2: Average total discounted reward. The results for `POMCP`, and `DESPOT` are taken from (Somani et al., 2013).

|  | $Tag$ | $LaserTag$ | $RS(15, 15)$ | $Pocman$ |
|---|---|---|---|---|
| `W-MCTS-OS` | $-6.05 \pm 0.56$ | $-18.17 \pm 0.46$ | $19.76 \pm 0.28$ | $297.98 \times 2.83$ |
| `W-MCTS-TS` | $-5.90 \pm 0.66$ | $-8.75 \pm 0.5$ | $20.29 \pm 0.22$ | $315.45 \pm 2.15$ |
| `POMCP` | $-7.14 \pm 0.28$ | $-19.58 \pm 0.06$ | $12.23 \pm 0.32$ | $294.16 \pm 4.06$ |
| `AB-DESPOT` | $-6.57 \pm 0.26$ | $-11.13 \pm 0.30$ | $18.18 \pm 0.30$ | $290.34 \pm 4.12$ |
| `AR-DESPOT` | $-6.26 \pm 0.28$ | $-9.34 \pm 0.26$ | $18.57 \pm 0.30$ | $307.96 \pm 4.22$ |

of $\alpha$-**Divergence.** We explored several values of $\alpha$ to vary how aggressively our backups shift between average-like and max-like behavior. When $\alpha$ approaches 0 or $\infty$, the update becomes nearly a pure average ($p = 1$) or nearly a max backup, respectively. In practice, we found that moderate $\alpha$ values often provide a suitable balance between these extremes, and we report results with the best-performing choices. Although a more extensive sensitivity analysis could be conducted, the core takeaway is that combining power-mean backups with variance propagation significantly enhances performance in highly stochastic tasks.

### 8.3. Key Performance Factors

The superior performance of our method stems from two complementary components that address fundamental lim-

itations in existing MCTS approaches for stochastic and partially observable environments:

**Explicit Variance Propagation.** Unlike previous methods that only propagate point estimates or use fixed variance models, our approach dynamically updates both means and variances at each node through the $L^1$-Wasserstein barycenter formulation. This capability is particularly crucial in highly stochastic and partially observable environments where uncertainty quantification directly impacts decision quality. Our experimental results demonstrate consistent improvements over Bayesian MCTS methods: we achieve up to 80% improvement over `DNG` in *Frozen-Lake*, and significant gains over `POMCP` across all POMDP environments, with particularly notable improvements of 55.31% in *LaserTag* and 65.90% in *rocksample*(15,15). Additionally, we observe improvements of up to 21.38% over `AB-DESPOT` in *LaserTag*, highlighting the effectiveness of our distributional approach.

**Flexibility in Balancing Exploration-Exploitation.** Our approach's ability to interpolate between average-like and max-like backups through the $\alpha$-divergence parameter allows adaptive behavior across varying levels of stochasticity. In highly stochastic environments such as *FrozenLake* and *NChain*, we found that moderate $\alpha$ values (leading to more average-like updates with $p$ closer to 1) performed optimally by preventing overestimation bias. Conversely, in environments with more deterministic regions of the state space, larger $\alpha$ values (yielding more max-like behavior) proved beneficial for faster convergence to optimal policies. This flexibility, combined with our Thompson sampling strategy, enables our algorithm to automatically adapt its exploration-exploitation balance based on the empirical variance observed at each node.

The synergy between these two components—principled uncertainty propagation and adaptive backup operators—explains why `W-MCTS` consistently outperforms both classical MCTS variants and existing Bayesian approaches across our diverse set of benchmark environments.

## 9. Conclusion

We proposed *Wasserstein MCTS*, an algorithm that models node values as Gaussian distributions and employs $L^1$-Wasserstein barycenters with $\alpha$-divergences to unify average- and max-like backups. Coupled with Thompson sampling or optimistic selection, our method achieves strong empirical performance while offering $\mathcal{O}(n^{-1/2})$ convergence guarantees. Experiments in both stochastic MDPs and POMDPs show significant improvements over classic baselines and Bayesian MCTS variants. Future work includes extending these Wasserstein-based ideas to open-loop planning (Leurent & Maillard, 2020; Bubeck &

Munos, 2010) for even broader applicability.

## Impact Statement

Our proposed *Wasserstein MCTS* algorithm offers a principled way to tackle complex, stochastic tasks in both fully and partially observable domains. Potential applications include robotics, autonomous systems, and large-scale resource management, all of which require adaptive planning strategies to handle real-world variability. While we do not anticipate immediate negative societal implications, responsible deployment remains essential. As with any AI-driven technology, understanding ethical, economic, and security ramifications—such as autonomy in safety-critical systems—should guide practical use.

### Acknowledgments

This research is funded by Hanoi University of Science and Technology (HUST) under Project No. T2024-TD-024, the French Ministry of Higher Education and Research, the Hautsde-France region, Inria, the MEL, the French National Research Agency under PEPR IA FOUNDRY project (ANR-23-PEIA-0003).

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

**Outline**

- Notations will be described in Section A.

- Hyperparameters are provided in Section B.

- Derivation of Wasserstein barycenter with Gaussian and particle filter distributions will be described in Section C.

- Supporting Lemmas will be provided in Section D.

- Full proof for the convergence of Wasserstein Non-stationary multi-armed bandit will be provided in Section E.

- Full proof for the convergence of Wasserstein Monte-Carlo tree search will be provided in Section F.

## A. Notations

Table 3: List of all notations of Wasserstein barycenter with Gaussian and particle filter distributions.

| Notation | Type | Description |
|---|---|---|
| $\mathcal{N}(m, \delta^2)$ | $\mathbb{R}$ | Gaussian distribution with mean $m$, standard deviation $\delta$ |
| $(\mathcal{X}, d)$ | | complete separable metric (Polish) space |
| $W_q(\mu, \nu)$ | | $L^q$-Wasserstein distance between $\mu, \nu$ |
| $W_1(\mu, \nu)$ | | $L^1$-Wasserstein distance between $\mu, \nu$ |
| $F_{p(x)}^{-1}(t)$ | | quantile function of a distribution $p(x)$ |
| $\Gamma(\mu, \nu)$ | $\mathcal{X} \times \mathcal{Y}$ | set of measures on $\mathcal{X} \times \mathcal{Y}$ with marginals $\mu, \nu$ |
| $d(X, Y)$ | $\mathbb{R}$ | distance between $X$ and $Y$ |
| $D_{f_\alpha}(X \| Y)$ | $\mathbb{R}$ | $\alpha$-divergence distance between $X$ and $Y$ |
| $\mathrm{erf}^{-1}(t)$ | | the inverse of the function $\frac{2}{\sqrt{\pi}} \int_0^t \exp\{-x^2\} dx$ |

## B. Experimental setup and Parameters selection

All the experiments were done on an Intel(R) Core(TM) i9-14900K 3.20 GHz 24 cores/CPU.

To compare the performance of `W-MCTS` to other state-of-the-art planning algorithms, we run several experiments on standard MDP as well as POMDP environments. For comparison, we consider `UCT` (Kocsis et al., 2006), `Power-UCT` (Dam et al., 2019), `DNG` (Bai et al., 2013) and `D2NG` (Bai et al., 2014). The hyperparameters are tuned using grid-search. Except for the case of *Pocman* environment, we scale the rewards into the range [0, 1]. We use the discount factor $\gamma = 0.95$. For `DNG`, `D2NG`, we set hyperparameters as recommended in the paper and from the author's source code (Bai et al., 2013; 2014). We set exploration constant for `UCT`, `Power-UCT` to $\sqrt{2}$. We set initial standard deviation value to $std = 30$. In all *Rocksample* and *Pocman* environments, we set the heuristic for rollouts as $treeknowledge = 0, rolloutknowledge = 1$. For all environments, we increase the value of $p$ and choose the best power mean $p$ value for `Power-UCT`, and `W-MCTS`. Details can be found in Table 6. For POMDP environments such as *Rocksample*, *Pocman* we get the source code released from the author of `DNG` (Bai et al., 2013) and `D2NG` (Bai et al., 2014)[1].

---

[1]https://github.com/aijunbai/thompson-sampling

Table 4: List of all notations of Wasserstein Non-stationary multi-armed bandits.

| Notation | Type | Description |
|:---:|:---:|:---:|
| $K$ | $\mathbb{N}$ | number of arms/actions |
| $\mu_k$ | $\mathbb{R}$ | mean value of arm $k$ |
| $\mu^*$ | $\mathbb{R}$ | optimal mean value |
| $\triangle_k$ | $\mathbb{R}$ | $\triangle_k = \mu^* - \mu_k$ |
| $\triangle$ | $\mathbb{R}$ | $\triangle = \max_{k \in [K]}\{\triangle_k\}$ |
| $\overline{X}_s^*$ | $\mathbb{R}$ | average reward of the optimal arm after $s$ visitations |
| $F_s^*$ | $\mathbb{R}$ | CDF of Gaussian with mean $\overline{X}_s^*$ |
| $T_k(n)$ | $\mathbb{N}$ | number of visitations of arm $k$ at timesteps $n$ |
| $\overline{X}_n(p)$ | $\mathbb{R}$ | power mean backup operator with power $p$ |
| $\overline{X}_{k,T_k(n)}$ | $\mathbb{R}$ | average rewards of arm $k$ after $T_k(n)$ visits |

## C. Derivation of Wasserstein barycenter with Gaussian and particle filter distributions

We revisit the definition of Wasserstein distance: The $L^q$-Wasserstein distance (with $q > 0$) between two distributions $\mu, \nu$ with the cost function $d(x,y) : \mathcal{X} \times \mathcal{Y} \to \mathbb{R}$ is defined as

$$W_q(\mu,\nu) = \left( \inf_{\rho \in \Gamma(\mu,\nu)} \mathbb{E}_{X,Y \sim \rho}[d(X,Y)^q] \right)^{1/q}, \tag{4}$$

here $\Gamma(\mu,\nu)$ is the set of measures on $\mathcal{X} \times \mathcal{Y}$ with marginals $\mu, \nu$.
Define $F_{p(x)}^{-1}(t)$ as the quantile function of a distribution

$$p(x) : F_{p(x)}^{-1}(t) = \inf\{x \in \mathbb{R}, t \leqslant F_p(x)\}. \tag{5}$$

With $d(X,Y) = |X - Y|$ as the Euclidean distance, we can derive

$$W_q^q(\mu,\nu) = \left( \int_0^1 |F_\mu^{-1}(t) - F_\nu^{-1}(t)|^q dt \right). \tag{6}$$

With $d(X,Y) = D_{f_\alpha}(X||Y)$, as the $\alpha$-divergence distance (defined in section 4.1), we can derive

$$W_q^q(\mu,\nu) = \left( \int_0^1 D_{f_\alpha}(F_\mu^{-1}(t)||F_\nu^{-1}(t))^q dt \right). \tag{7}$$

### C.1. $L^1$-Wasserstein barycenter with $\alpha$-divergence distance

We have

$$W_1(\mu,\nu) = \inf_{\rho \in \Gamma(\mu,\nu)} \mathbb{E}_{X,Y \sim \rho}[d(X,Y)] = \inf_{\rho \in \Gamma(\mu,\nu)} \mathbb{E}_{X,Y \sim \rho}[D_{f_\alpha}(X,Y)]. \tag{8}$$

Table 5: List of all notations of Wasserstein Monte-Carlo Tree Search.

| Notation | Type | Description |
|---|---|---|
| KL | | KL divergence |
| $V_m(s_h)$ | $\mathbb{R}$ | optimal mean of V value at root state $s_h$, at depth $(h)$ |
| $Q_m(s_h, a_k)$ | $\mathbb{R}$ | mean of Q value function at state $s_h$, action $a_k$, at depth $(h)$ |
| $\overline{V}_m(s_h, n)$ | $\mathbb{R}$ | empirical estimated mean of V value at state $s_h$ after n visitations at depth $(h)$ |
| $\overline{Q}_m(s_h, a_k, n)$ | $\mathbb{R}$ | empirical estimated mean of Q value at root at state $s_h$, action $a_k$ after n visitations at depth $(h)$ |
| $V_m(s_h)$ | $\mathbb{R}$ | optimal mean of V value at depth $(h)$ at state $s_h$ |
| $Q_m(s_h, a_k)$ | $\mathbb{R}$ | mean of Q value function at depth $(h)$ at state $s_h$, action $a_k$ |
| $\overline{V}_m(s_h, n)$ | $\mathbb{R}$ | empirical estimated mean of V value at depth $(h)$ at state $s_h$ after n visitations |
| $\overline{Q}_m(s_h, a_k, n)$ | $\mathbb{R}$ | empirical estimated mean of Q value at depth $(h)$ at state $s_h$, action $a_k$ after n visitations |
| $T_{s_h, a_k}(n)$ | $\mathcal{N}$ | number of plays of action $a_k$ at state $s_h$ at timestep $n$ |
| $T_{s, a_k}^{s'}(n)$ | $\mathcal{N}$ | number of plays of action $a_k$ at state $s$ to state $s'$ at timestep $n$ |

Table 6: List of all hyperparameters.

| Environments | p Value Search | Best p Value |
|---|---|---|
| *FrozenLake* | $p = 1, 2, 4, 10, 100$ | W-MCTS-OS (p=100),W-MCTS-TS (p=100),Power-UCT(p=100) |
| *NChain* | $p = 1, 2, 4, 8, 15, 100$ | W-MCTS-OS (p=4),W-MCTS-TS (p=100),Power-UCT(p=8) |
| *RiverSwim* | $p = 1, 2, 4, 8, 15, 100$ | W-MCTS-OS (p=100),W-MCTS-TS (p=15),Power-UCT(p=15) |
| *SixArms* | $p = 1, 2, 4, 8, 15, 100$ | W-MCTS-OS (p=100),W-MCTS-TS (p=100),Power-UCT(p=8) |
| *Taxi* | $p = 1, 2, 4, 8, 15, 100$ | W-MCTS-OS (p=15),W-MCTS-TS (p=15),Power-UCT(p=15) |
| *Rocksample(11x11)* | $p = 10, 50, 80, 100, 150$ | W-MCTS-OS (p=150),W-MCTS-TS (p=100) |
| *Rocksample(15x15)* | $p = 10, 50, 80, 100, 150$ | W-MCTS-OS (p=100),W-MCTS-TS (p=100) |
| *Rocksample(15x35)* | $p = 10, 80, 100$ | W-MCTS-OS (p=150),W-MCTS-TS (p=10) |
| *Pocman* | $p = 1, 2, 4, 8, 10, 100$ | W-MCTS-OS (p=1),W-MCTS-TS (p=100) |

We find the lower bound of $W_1(\mu, \nu)$ with $\alpha$-divergence as a measure cost function.

Let denote $\mathcal{N}(m, \delta^2)$ as a Gaussian distribution with mean $m$ and standard deviation $\delta$. With $\mu = \mathcal{N}(m_1, \delta_1^2), \nu = \mathcal{N}(m_2, \delta_2^2)$ We first want to show that by applying Data Processing Inequalities (Lemma 2.1 (Gerchinovitz et al., 2020)),

with $h(X) = X - m_1$, and $g(X) = X - m_2$, we can derive

$$W_1(\mu, \nu) = \inf_{\rho \in \Gamma(\mu, \nu)} \mathbb{E}_{X, Y \sim \rho}[D_{f_\alpha}(X, Y)]] \geqslant \inf_{\rho \in \Gamma(\mu, \nu)} \mathbb{E}_{X, Y \sim \rho}[D_{f_\alpha}(X - m_1, Y - m_1)]$$

$$= W_1(\mathcal{N}(0, \delta_1^2), \mathcal{N}(m_2 - m_1, \delta_2^2)), \tag{9}$$

and

$$W_1(\mu, \nu) = \inf_{\rho \in \Gamma(\mu, \nu)} \mathbb{E}_{X, Y \sim \rho}[D_{f_\alpha}(X, Y)]] \geqslant \inf_{\rho \in \Gamma(\mu, \nu)} \mathbb{E}_{X, Y \sim \rho}[D_{f_\alpha}(X - m_2, Y - m_2)]$$

$$\geqslant \inf_{\rho \in \Gamma(\mu, \nu)} \mathbb{E}_{X, Y \sim \rho}[D_{f_\alpha}(m_2 - X, m_2 - Y)](\text{ with the transform function } f(X) = -X)$$

$$= W_1(\mathcal{N}(m_2 - m_1, \delta_1^2), \mathcal{N}(0, \delta_2^2)). \tag{10}$$

Now according to (7), the $L^1$-Wasserstein distance with $\alpha$-divergence distance is defined as

$$W_1(\mu, \nu) = \left( \int_0^1 D_{f_\alpha}(F_\mu^{-1}(t) || F_\nu^{-1}(t)) dt \right). \tag{11}$$

We show that the quantile function of a Gaussian distribution (Soch, 2020) $F = \mathcal{N}(\mu, \delta^2)$ is

$$F^{-1}(t) = \sqrt{2}\delta \text{erf}^{-1}(2t - 1) + \mu, \tag{12}$$

where $\text{erf}^{-1}(t)$ is the inverse of the function $\frac{2}{\sqrt{\pi}} \int_0^t \exp\{-x^2\} dx$.

Therefore, the $L^1$-Wasserstein distance with $\alpha$-divergence distance as the cost function between two Gaussian distributions $\mu = \mathcal{N}(m_1, \delta_1^2), \nu = \mathcal{N}(m_2, \delta_2^2)$ can be measured as

$$W_1(\mu, \nu) = \left( \int_0^1 D_{f_\alpha}(\sqrt{2}\delta_1 \text{erf}^{-1}(2t - 1) + m_1 || \sqrt{2}\delta_2 \text{erf}^{-1}(2t - 1) + m_2) dt \right).$$

Applying the convexity properties of $\alpha$-divergence (Cichocki & Amari, 2010), and from (9),(10) we have

$$W_1(\mu, \nu) \geqslant \frac{1}{2} \left( \int_0^1 D_{f_\alpha}(\sqrt{2}\delta_1 \text{erf}^{-1}(2t - 1) || \sqrt{2}\delta_2 \text{erf}^{-1}(2t - 1) + m_2 - m_1) dt \right.$$

$$\left. + \int_0^1 D_{f_\alpha}(\sqrt{2}\delta_1 \text{erf}^{-1}(2t - 1) + m_2 - m_1 || \sqrt{2}\delta_2 \text{erf}^{-1}(2t - 1)) dt \right)$$

$$\geqslant \left( \int_0^1 D_{f_\alpha}(\sqrt{2}\delta_1 \text{erf}^{-1}(2t - 1) + \frac{m_2 - m_1}{2} || \sqrt{2}\delta_2 \text{erf}^{-1}(2t - 1) + \frac{m_2 - m_1}{2}) \right)$$

$$= W_1(\mathcal{N}(\frac{m_2 - m_1}{2}, \delta_1^2), \mathcal{N}(\frac{m_2 - m_1}{2}, \delta_2^2)).$$

Applying Data Processing Inequalities (Lemma 2.1 (Gerchinovitz et al., 2020)), with $h(X) = X - \frac{m_2 - m_1}{2}$, we can derive

$$W_1(\mu, \nu) \geqslant W_1(\mathcal{N}(0, \delta_1^2), \mathcal{N}(0, \delta_2^2)) = \left( \int_0^1 D_{f_\alpha}(\sqrt{2}\delta_1 \text{erf}^{-1}(2t - 1) || \sqrt{2}\delta_2 \text{erf}^{-1}(2t - 1)) dt \right).$$

Let us consider the sequences $0 = t_0 \leqslant t_1 \leqslant ... \leqslant t_N = 1$, there exists $\xi_i \in [t_i, t_{i+1}]$ that

$$W_1(\mu, \nu) \geqslant \sum_{i=0}^{i=N}(t_{i+1} - t_i) D_{f_\alpha}(\sqrt{2}\delta_1 \text{erf}^{-1}(2\xi_i - 1) || \sqrt{2}\delta_2 \text{erf}^{-1}(2\xi_i - 1))$$

$$= \sum_{i=0}^{i=N} \Delta_i D_{f_\alpha}(\sqrt{2}\delta_1 \text{erf}^{-1}(2\xi_i - 1) || \sqrt{2}\delta_2 \text{erf}^{-1}(2\xi_i - 1)),$$

with $\Delta_i = (t_{i+1} - t_i)$. Since $D_{f_\alpha}(cP || cQ) = D_{f_\alpha}(P || Q)$ where $c$ is a constant. We can derive

$$W_1(\mu, \nu) \geqslant \sum_{i=0}^{i=N} \Delta_i D_{f_\alpha}(\delta_1 || \delta_2) = D_{f_\alpha}(\delta_1 || \delta_2). \tag{13}$$

We start with the first Proposition about the closed solutions of mean and variance of a Gaussian value function $\mathcal{V}(s)$ as V-posterior $L^1$-Wasserstein barycenter of all action value function distributions $\mathcal{Q}(s, a)$.

**Proposition 1.** *Consider the V-posterior value function $\mathcal{V}(s)$ as a Gaussian: $\mathcal{N}(\overline{m}(s), \overline{\delta}^2(s))$. Let's define each $\mathcal{Q}(s, a)$ as the Q function child node of $\mathcal{V}(s)$. Each $\mathcal{Q}(s, a)$ is assumed as a Gaussian distributions $\mathcal{Q}(s, a) : \mathcal{N}(m(s, a), \delta(s, a)^2)$. If the value function $\mathcal{V}(s)$ is defined as the Wasserstein barycenter of the Q function $\mathcal{Q}(s, a)$ given the policy $\bar{\pi}$, we will have:*

$$\overline{m}(s) = (\mathbb{E}_{a \sim \bar{\pi}}[m(s, a)^p])^{\frac{1}{p}} \tag{14}$$

$$\overline{\delta}(s) = (\mathbb{E}_{a \sim \bar{\pi}}[\delta(s, a)^p])^{\frac{1}{p}}, \tag{15}$$

*with $p = 1 - \alpha$.*

*Proof.* By the definition of the V-posterior value function, we have:

$$(\overline{\mu}(s), \overline{\delta}(s)) = \arg\min_{\mu, \delta} \left\{ \mathbb{E}_{\bar{\pi}}[W_1(\mathcal{V}(s) || \mathcal{Q}(s, a))] \right\}. \tag{16}$$

We first compute the standard deviation $\overline{\delta}(s)$.
From (13), and (16), we want to find $\overline{\delta}(s)$ that is the minimizer of

$$\overline{\delta}(s) = \arg\min_{\delta(s)} \left\{ \mathbb{E}_{\bar{\pi}}[D_{f_\alpha}(\delta(s) || \delta(s, a))] \right\}.$$

we derive $\overline{\delta}(s)$ is the solution of

$$\frac{\nabla \mathbb{E}_{a \sim \bar{\pi}}[D_{f_\alpha}(\delta(s) || \delta(s, a))]}{\nabla \delta(s)} = 0. \tag{17}$$

Since

$$\frac{\nabla f_\alpha(x)}{\nabla x} = \frac{\alpha(x^{\alpha-1} - 1)}{\alpha(\alpha - 1)} = \frac{x^{\alpha-1} - 1}{\alpha - 1}. \tag{18}$$

With $D_{f_\alpha}(x || y) = \sum_y y f_\alpha(\frac{x}{y})$, we can have

$$\frac{\nabla D_{f_\alpha}(x || y)}{\nabla x} = \sum_y \frac{(\frac{x}{y})^{\alpha-1} - 1}{\alpha - 1}. \tag{19}$$

We can derive

$$\mathbb{E}_{a \sim \bar{\pi}} \left[ \frac{(\frac{\overline{\delta}(s)}{\delta(s,a)})^{\alpha-1} - 1}{(\alpha - 1)} \right] = 0 \implies \mathbb{E}_{a \sim \bar{\pi}} \left[ (\frac{\overline{\delta}(s)}{\delta(s, a)})^{\alpha-1} - 1 \right] = 0. \tag{20}$$

Now we can define $p = 1 - \alpha$ that leads to

$$\overline{\delta}(s) = (\mathbb{E}_{a \sim \bar{\pi}}[\delta(s, a)^p])^{\frac{1}{p}}. \tag{21}$$

To compute $\bar{\mu}(s)$. Let's revisit here again the definition of $L^1-$Wasserstein distance between two Gaussian distributions $\mu(m_1, \delta_1^2), \nu(m_2, \delta_2^2)$.

$$W_1(\mu, \nu) = \inf\{\mathbb{E}[D_{f_\alpha}(\mu || \nu)]\}. \tag{22}$$

According to Jensen's inequality(Perlman, 1974) we can derive

$$\mathbb{E}[D_{f_\alpha}(\mu || \nu)] \geqslant D_{f_\alpha}(\mathbb{E}[\mu] || \mathbb{E}[\nu]) = D_{f_\alpha}(m_1 || m_2). \tag{23}$$

Therefore, according to the definition of Wasserstein barycenter, the mean of a Gaussian V-posterior value function $\mathcal{V}(s)$ can be derived as

$$\overline{m}(s) = \arg\min_{m(s)} \mathbb{E}_{a \sim \bar{\pi}}[D_{f_\alpha}(m(s) || m(s, a))]. \tag{24}$$

Following the same steps as to compute $\delta(s)$, we can get

$$\overline{m}(s) = (\mathbb{E}_{a \sim \bar{\pi}}[m(s, a)^p])^{\frac{1}{p}}, \tag{25}$$

with $p = 1 - \alpha$ that concludes the proof.

$\square$

Next, we consider each node as an equally weighted Particle model and derive the following proposition.

**Proposition 2.** *Let's assume the V-posterior value function $\mathcal{V}(s)$ as a equally weighted Particle model: $\overline{x_i}(s) : i \in [1, M]$. $M$ is an integer and $M \geqslant 1$. Let's assume each Q function $\mathcal{Q}(s, a)$ has M particles $x_i(s, a), i \in [1, M]$. If the value function $\mathcal{V}(s)$ is defined as the Wasserstein barycenter of the Q function $\mathcal{Q}(s, a)$ given the policy $\bar{\pi}$, each particle $(\overline{x_i}(s), i \in [1, M])$ can be estimated as*

$$\overline{x_i}(s) = (\mathbb{E}_{a \sim \bar{\pi}}[x_i(s, a)^p])^{1/p}, \tag{26}$$

*with $p = 1 - \alpha$.*

*Proof.* We can compute the quantile function of $\mu$ and $\nu$ as

$$F_\mu^{-1}(t) = \sum_{i=1}^{M} x_i \mathbf{1}_{I_i}(t), F_\nu^{-1}(t) = \sum_{i=1}^{M} y_i \mathbf{1}_{I_i}(t). \tag{27}$$

Therefore from (11) we can get

$$W_1(\mu, \nu) = \left( \int_0^1 D_{f_\alpha}(F_\mu^{-1}(t) || F_\nu^{-1}(t)) dt \right) \tag{28}$$

$$= \sum_{i=1}^{M} \left( \int_{I_i} D_{f_\alpha}(F_\mu^{-1}(t) || F_\nu^{-1}(t)) dt \right) \tag{29}$$

$$= \sum_{i=1}^{M} \left( \int_{I_i} D_{f_\alpha}(x_i || y_i) dt \right) \tag{30}$$

$$= \sum_{i=1}^{M} D_{f_\alpha}(x_i || y_i) \left( \int_{I_i} dt \right) \tag{31}$$

$$= \sum_{i=1}^{M} w_i D_{f_\alpha}(x_i || y_i). \tag{32}$$

We can see that for each particle $(\overline{x_i}(s), i \in [1, M])$, we can derive

$$\overline{x_i}(s) = \underset{x_i(s)}{\arg \min} \, \mathbb{E}_{a \sim \bar{\pi}}[D_{f_\alpha}(x_i(s) || x_i(s, a))] \tag{33}$$

$$\implies \overline{x_i}(s) = (\mathbb{E}_{a \sim \bar{\pi}}[x_i(s, a)^p])^{1/p}, \tag{34}$$

with $p = 1 - \alpha$. $\qquad\square$

## D. Supporting Lemmas

We will make use of the following basic results.

**Lemma 1.** *(Minkowski's inequality) Given $p \geqslant 1, \{x_i, y_i\} \in \mathbb{R}, i = 1, 2, ..., n$, then we have the following inequality*

$$\left( \sum_i (|x_i + y_i|)^p \right)^{\frac{1}{p}} \leqslant \left( \sum_i (|x_i|)^p \right)^{\frac{1}{p}} + \left( \sum_i (|y_i|)^p \right)^{\frac{1}{p}}. \tag{35}$$

*Proof.* This is a basic result. $\qquad\square$

**Lemma 2.** *(Markov's inequality) If $X$ is a nonnegative random variable and $a > 0$, then the probability that X is at least a is at most the expectation of X divided by a:*

$$\mathbf{Pr}(X > a) \leqslant \frac{\mathbb{E}[X]}{a}. \tag{36}$$

# E. Convergence of Wasserstein Non-stationary multi-armed bandits

We note that in an MCTS tree, each node is considered a non-stationary multi-armed bandit where the average mean drifts due to the given action selection strategy. Therefore, we first study the convergence of Wasserstein non-stationary multi-armed bandits where the action selection is Thompson sampling, with the power mean backup operator at the root node. Detailed descriptions of the Wasserstein Non-stationary multi-armed bandits settings can be found in the main article in the Theoretical Analysis section.

We briefly summarize the theoretical results below. Lemma 6 is about the upper bound on the expectation of the number of suboptimal arms playing, following the corresponding Theorem 4.2 in (Jin et al., 2022). Lemma 7 is about the bias of the expected value of the power mean backup operator, which follows the result as Theorem 1 in Stochastic-Power-UCT (Dam et al., 2024b). Theorem 2 deals with the polynomial concentration of the power mean backup operator around the optimal mean at the root node of the non-stationary Wasserstein problem for multi-armed bandits. This theorem plays an important role in deriving the polynomial convergence of the choice of the optimal action at the root node in the Wasserstein MCTS tree, described in the next section.

Now, we will find an upper bound for the expectation of numbers of pulling a suboptimal arm. Let us define the event $E_{k,\varepsilon}(t) = \{\theta_k(t) \leqslant \mu^* - \varepsilon\}$ for all $k \in [K], \varepsilon > 0, \theta_k(t)$ is sampled from $\mathcal{N}(\overline{X}_k, V/T_k(n))$ at timestep $t$. Let us consider the decomposition

$$\mathbb{E}[T_k(n)] = 1 + \mathbb{E}\Big[ \sum_{t=K+1}^{n} \mathbf{1}\{A_t = a_k, E_{k,\varepsilon}(t)\} + \sum_{t=K+1}^{n} \mathbf{1}\{A_t = a_k, E_{k,\varepsilon}^c(t)\}\Big] \tag{37}$$

$$= 1 + \underbrace{\mathbb{E}\Big[ \sum_{t=K+1}^{n} \mathbf{1}\{A_t = a_k, E_{k,\varepsilon}(t)\}\Big]}_{A} + \underbrace{\mathbb{E}\Big[ \sum_{t=K+1}^{n} \mathbf{1}\{A_t = a_k, E_{k,\varepsilon}^c(t)\}\Big]}_{B}. \tag{38}$$

Here $E^c$ is the complement of an event $E$, $\varepsilon > 8\sqrt{V/n}$ is an arbitrary constant.

**Bounding Term A:** Let's define

$$\alpha_s = \sup_{x \in [0, \mu^* - \varepsilon)} \Big\{ \mathrm{KL}(\mu^* - \varepsilon - x, \mu^*) \leqslant 4\log(\frac{n}{s})/s \Big\}. \tag{39}$$

**Lemma 3.** *(Lemma A.1 (Jin et al., 2022)) Let $M = \lceil 16V \log(n\varepsilon^2/V)/\varepsilon^2 \rceil$, and $\alpha_s$ be the same as defined in (39) then*

$$\mathbb{E}\Big[ \sum_{t=K+1}^{n} \mathbf{1}\{A_t = a_k, E_{k,\varepsilon}(t)\}\Big] \leqslant \sum_{s=1}^{M} \mathbb{E}\Big[\Big(\frac{1}{1 - F_s^*(\mu^* - \varepsilon)} - 1\Big).\mathbf{1}\{\overline{X}_s^* \in (\mu^* - \varepsilon - \alpha_s, 1]\}\Big] + \ominus\Big(\frac{V}{\varepsilon^2}\Big), \tag{40}$$

*where $F_s^*$ is the CDF of Gaussian with mean $\overline{X}_s^*$, $\overline{X}_s^*$ is the average reward of the optimal arm after $s$ visitations.*

**Lemma 4.** *(Lemma A.2 (Jin et al., 2022)) Let $M = \lceil 16V \log(n\varepsilon^2/V)/\varepsilon^2 \rceil$. Then*

$$\sum_{s=1}^{M} \mathbb{E}_{\overline{X}_s^*}\Big[\Big(\frac{1}{1 - F_s^*(\mu^* - \varepsilon)}\Big).\mathbf{1}\{\overline{X}_s^* \in (\mu^* - \varepsilon - \alpha_s, 1]\}\Big] = \Theta\Big(\frac{V \log(n\varepsilon^2/V)}{\varepsilon^2}\Big). \tag{41}$$

**Bounding Term B:**

**Lemma 5.** *(Lemma C.1 (Jin et al., 2022)) Let $N = \min\{\frac{1}{1 - \frac{\mathrm{KL}(\mu_k + \rho_k, \mu^* - \varepsilon)}{\log(n\varepsilon^2/V)}}, 2\}$. For any $\rho_k, \varepsilon > 0$ that satisfies $\varepsilon + \rho_k < \Delta_i$,*

*then*

$$\mathbb{E}\Big[ \sum_{t=K+1}^{n} \mathbf{1}\{A_t = k, E_{k,\varepsilon}^c(t)\}\Big] \leqslant 1 + \frac{2V}{\rho_k^2} + \frac{V}{\varepsilon^2} + \frac{N \log(n\varepsilon^2/V)}{KL(\mu_k + \rho_k, \mu^* - \varepsilon)}. \tag{42}$$

From Assumption 1, we derive the upper bound for the expectation of the number of plays of a suboptimal arm.

**Lemma 6.** *Consider Thompson Sampling strategy (using power mean estimator) applied to a non-stationary problem where the pay-off sequence satisfies Assumption 1. Fix $\varepsilon \geqslant 0$. Let $T_k(n)$ denote the number of plays of arm $k$. Then if $k$ is the index of a suboptimal arm, then each sub-optimal arm $k$ is played in expectation at most*

$$\mathbb{E}[T_k(n)] \leqslant \Theta\left(1 + \frac{V \log(n\Delta_k^2/V)}{\Delta_k^2}\right). \tag{43}$$

*Proof.* The proof of Lemma 6 closely follows Theorem 4.2((Jin et al., 2022)) by observing results from Lemma 3, 4, 5. From equation 38, putting all Lemma 3, 4, 5, we have

$$\mathbb{E}[T_k(n)] = \Theta\left(1 + \frac{V \log(n\varepsilon^2/V)}{(\Delta_k - \varepsilon - \rho_k)^2} + \frac{V}{\rho_k^2} + \frac{V \log(n\varepsilon^2/V)}{\varepsilon^2}\right). \tag{44}$$

Set $\varepsilon = \rho_k = \Delta_k/4$, we derive

$$\mathbb{E}[T_k(n)] \leqslant \Theta\left(1 + \frac{V \log(n\Delta_k^2/V)}{\Delta_k^2}\right). \tag{45}$$

$\square$

**Lemma 7.** *Consider a non-stationary problem where the pay-off sequence satisfies Assumption 1. We consider a bandit algorithm that selects each arm as*

$$a = \underset{a_i, i \in \{1...K\}}{\arg\max} \{\theta_i \sim \mathcal{N}(\overline{X}_{k,n}, V/T_k(n))\}.$$

*Let us define the power mean estimator $\overline{X}_n(p)$ as $\overline{X}_n(p) = \left(\sum_{a=1}^{K} \frac{T_a(n)}{n} \overline{X}_{a,T_a(n)}^p\right)^{\frac{1}{p}}$, and $\delta_{\star,n} = \mu_\star - \mu_{\star,n}$ For any $p \geqslant 1, \varepsilon_0 > 0$, we have*

$$\left|\mathbb{E}[\overline{X}_n(p)] - \mu_\star\right| \leqslant |\delta_{\star,n}| + \frac{R}{n} \sum_{a=1, a \neq a_*}^{K} \Theta\left(1 + \frac{V \log(n\Delta_k^2/V)}{\Delta_k^2}\right) \tag{46}$$

*Proof.* We observe that

$$\left|\overline{X}_n(p) - \mu_\star\right| \leqslant \left|\overline{X}_n(p) - \mu_{\star,n}\right| + |\mu_\star - \mu_{\star,n}| = \left|\overline{X}_n(p) - \mu_{\star,n}\right| + |\delta_{\star,n}| \tag{47}$$

Furthermore,

$$\overline{X}_{a,T_a(n)} \leqslant \mu_{a,n} + \left|\overline{X}_{a,T_a(n)} - \mu_{a,n}\right|. \tag{48}$$

Since $\mu_{\star,n} = \max_{a \in [K]}\{\mu_{a,n}\}$, we have

$$\overline{X}_n(p) - \mu_{\star,n} = \overline{X}_n(p) - \sum_{a=1}^{K} T_a(n)\mu_{\star,n} \leqslant \left(\sum_{a=1}^{K} \frac{T_a(n)}{n}\left(\overline{X}_{a,T_a(n)}\right)^p\right)^{\frac{1}{p}} - \left(\sum_{a=1}^{K} \frac{T_a(n)}{n}\left(\mu_{a,n}\right)^p\right)^{\frac{1}{p}} \tag{49}$$

$$= \frac{\left(\sum_{a=1}^{K} T_a(n)\left(\overline{X}_{a,T_a(n)}\right)^p\right)^{\frac{1}{p}} - \left(\sum_{a=1}^{K} T_a(n)\left(\mu_{a,n}\right)^p\right)^{\frac{1}{p}}}{n^{\frac{1}{p}}} \tag{50}$$

Applying Minkowski's inequality from Lemma 1, and the result of equation 48, we have

$$\overline{X}_n(p) - \mu_{\star,n} \leqslant \frac{\left(\sum_{a=1}^{K} T_a(n)\left(\mu_a + \left|\overline{X}_{a,T_a(n)} - \mu_{a,n}\right|\right)^p\right)^{\frac{1}{p}} - \left(\sum_{a=1}^{K} T_a(n)\left(\mu_{a,n}\right)^p\right)^{\frac{1}{p}}}{n^{\frac{1}{p}}} \tag{51}$$

$$\leqslant \frac{\left(\sum_{a=1}^{K} T_a(n)\left(\left|\overline{X}_{a,T_a(n)} - \mu_{a,n}\right|\right)^p\right)^{\frac{1}{p}}}{n^{\frac{1}{p}}} \tag{52}$$

On the other hand,

$$\mu_{\star,n} - \overline{X}_n(p) = \frac{n\mu_{\star,n} - n\overline{X}_n(p)}{n} = \frac{n\mu_{\star,n} - (\sum_{a=1}^{K} T_a(n)\mu_{a,n}) + \sum_{a=1}^{K} T_a(n)\mu_{a,n} - n\overline{X}_n(p)}{n} \tag{53}$$

$$= \frac{\sum_{a=1,a\neq a_*}^{K} T_a(n)\left|\mu_{\star,n} - \mu_{a,n}\right| + \sum_{a=1}^{K} T_a(n)\mu_{a,n} - n\overline{X}_n(p)}{n} \tag{54}$$

$$\leqslant R \sum_{a=1,a\neq a_*}^{K} \frac{T_a(n)}{n} + \sum_{a=1}^{K} \frac{T_a(n)}{n}\mu_{a,n} - \overline{X}_n(p) \tag{55}$$

Because power mean is an increasing function of $p$, so that

$$\sum_{a=1}^{K} \frac{T_a(n)}{n}\mu_{a,n} \leqslant \left(\sum_{a=1}^{K} \frac{T_a(n)}{n}\left(\mu_{a,n}\right)^p\right)^{1/p}.$$

Furthermore, we observe that

$$\mu_{a,n} \leqslant \overline{X}_{a,T_a(n)} + \left|\overline{X}_{a,T_a(n)} - \mu_{a,n}\right|.$$

So that, from equation 55 we have

$$\mu_{\star,n} - \overline{X}_n(p) \leqslant R \sum_{a=1,a\neq a_*}^{K} \frac{T_a(n)}{n} + \left(\sum_{a=1}^{K} \frac{T_a(n)}{n}\left(\mu_{a,n}\right)^p\right)^{1/p} - \overline{X}_n(p) \tag{56}$$

$$\leqslant R \sum_{a=1,a\neq a_*}^{K} \frac{T_a(n)}{n} \tag{57}$$

$$+ \frac{\left(\sum_{a=1}^{K} T_a(n)\left(\overline{X}_{a,T_a(n)} + \left|\overline{X}_{a,T_a(n)} - \mu_{a,n}\right|\right)^p\right)^{\frac{1}{p}} - \left(\sum_{a=1}^{K} T_a(n)\left(\overline{X}_{a,T_a(n)}\right)^p\right)^{\frac{1}{p}}}{n^{\frac{1}{p}}} \tag{58}$$

$$\overset{\text{(Minkovski's inequality)}}{\leqslant} R \sum_{a=1,a\neq a_*}^{K} \frac{T_a(n)}{n} + \frac{\left(\sum_{a=1}^{K} T_a(n)\left(\left|\overline{X}_{a,T_a(n)} - \mu_{a,n}\right|\right)^p\right)^{\frac{1}{p}}}{n^{\frac{1}{p}}} \tag{59}$$

$$\overset{\text{(Properties of } L^p \text{ norm)}}{\leqslant} R \sum_{a=1,a\neq a_*}^{K} \frac{T_a(n)}{n} + \frac{\left(\sum_{a=1}^{K} T_a(n)\left(\left|\overline{X}_{a,T_a(n)} - \mu_{a,n}\right|\right)\right)}{n^{\frac{1}{p}}} \tag{60}$$

$$= R \sum_{a=1,a\neq a_*}^{K} \frac{T_a(n)}{n} + \frac{\sum_{a=1}^{K} \left(\left|\sum_{t}^{T_a(n)} X_{a,t} - T_a(n)\mu_{a,n}\right|\right)}{n^{\frac{1}{p}}} \tag{61}$$

Therefore

$$\left|\mathbb{E}[\overline{X}_n(p)] - \mu_{\star,n}\right| \leqslant R \sum_{a=1,a\neq a_*}^{K} \frac{\mathbb{E}[T_a(n)]}{n} + \frac{\mathbb{E}\left[\left(\left|\sum_{a=1}^{K}\sum_{t}^{T_a(n)} X_{a,t} - T_a(n)\mu_{a,n}\right|\right)\right]}{n^{\frac{1}{p}}} \tag{62}$$

$$= R \sum_{a=1,a\neq a_*}^{K} \frac{\mathbb{E}[T_a(n)]}{n} \tag{63}$$

Please note that because we study non-stationary bandits, $\mathbb{E}[\sum_{t}^{n} X_{a,t}] = n\mu_{a,n}$, therefore,

$$\frac{\mathbb{E}\left[\left(\left|\sum_{a=1}^{K}\sum_{t}^{T_a(n)} X_{a,t} - T_a(n)\mu_{a,n}\right|\right)\right]}{n^{\frac{1}{p}}} = 0$$

According to Lemma 7, we have

$$\left|\mathbb{E}[\overline{X}_n(p) - \mu_{\star,n}]\right| \leqslant |\delta_{\star,n}| + R \sum_{a=1,a\neq a_*}^{K} \frac{\mathbb{E}[T_a(n)]}{n} \leqslant |\delta_{\star,n}| + \frac{R}{n} \sum_{a=1,a\neq a_*}^{K} \Theta\left(1 + \frac{V\log(n\Delta_k^2/V)}{\Delta_k^2}\right), \qquad (64)$$

which concludes the proof. $\qquad\qquad\qquad\qquad\qquad\qquad\qquad\qquad\qquad\qquad\qquad\qquad\qquad\qquad\qquad\qquad\qquad\qquad$ □

**Theorem 2.** *For $a \in [K]$, let $(\overline{X}_{a,n})_{n\geqslant 1}$ be a sequence of estimator satisfying Assumption 1 and let $\mu_{\star} = \max_a\{\mu_a\}$. Assume that all the estimators are bounded in $[0, R]$. We consider a bandit algorithm that selects each arm as*

$$a = \operatorname*{argmax}_{a_i, i\in\{1...K\}} \{\theta_i \sim \mathcal{N}(\overline{X}_{k,n}, V/T_k(n))\}.$$

*Then, for all $p \in [1, \infty)$, the sequence of estimators*

$$\overline{X}_n(p) = \left(\sum_{a=1}^{K} \frac{T_a(n)}{n} \overline{X}_{a,T_a(n)}^p\right)^{\frac{1}{p}},$$

*where $T_a(n) = \sum_{t=1}^{n-1} \mathbb{1}(a_t = a)$ is the number of selections of $a$ prior to round $n$ satisfies*

$$\mathbf{Pr}\left(\left|\overline{X}_n(p) - \mu_{\star}\right| \geqslant \varepsilon\right) \leqslant Cn^{-1}\varepsilon^{-2}.$$

*Proof.* We first prove that $\lim_{n\to\infty} \mathbb{E}[\overline{X}_n(p)] = \mu_*$. According to the result of Lemma 7, we have

$$\left|\mathbb{E}[\overline{X}_n(p)] - \mu_{\star}\right| \leqslant |\delta_{\star,n}| + R \sum_{a=1,a\neq a_*}^{K} \frac{\mathbb{E}[T_a(n)]}{n} \qquad\qquad\qquad\qquad\qquad (65)$$

$$\leqslant |\delta_{\star,n}| + \frac{R}{n} \sum_{a=1,a\neq a_*}^{K} \left\{\frac{(1+\varepsilon_0)\log n}{\mathcal{K}^{(N)}(F_a, \mu_{\star})} + o(\log n) + O(1)\right\} \qquad (66)$$

with $\delta_{\star,n} = \mu_{\star} - \mu_{\star,n}$, and because $\lim_{n\to\infty} \mu_{*,n} = \mu_{\star}$, we can concludes that

$$\lim_{n\to\infty} \mathbb{E}[\overline{X}_n(p)] = \mu_*.$$

Second, we prove that

$$\forall n \geqslant 1, \forall \varepsilon > 0, \exists c > 0 \text{ that } \mathbb{P}\left(|\overline{X}_n(p) - \mu_{\star}| > \varepsilon\right) \leqslant cn^{-1}\varepsilon^{-2}.$$

We observe that

$$\left|\overline{X}_n(p) - \mu_{\star}\right| \leqslant \left|\overline{X}_n(p) - \mu_{\star,n}\right| + |\mu_{\star} - \mu_{\star,n}| = \left|\overline{X}_n(p) - \mu_{\star,n}\right| + |\delta_{\star,n}| \qquad (67)$$

$$\Longrightarrow \mathbb{P}(|\overline{X}_n(p) - \mu_{\star}| \geqslant \varepsilon) \leqslant \mathbb{P}(|\overline{X}_n(p) - \mu_{\star,n}| \geqslant \varepsilon/2) + \mathbb{P}(|\delta_{\star,n}| \geqslant \varepsilon/2). \qquad (68)$$

Because $\lim_{n\to n} |\delta_{\star,n}| = 0$, therefore, $\exists N_0 > 0$ such that $\forall n \geqslant N_0$, we have $|\delta_{\star,n}| < \varepsilon/2$ that means

$$\forall n > N_0, \mathbb{P}(|\delta_{\star,n}| \geqslant \varepsilon/2) = 0.$$

Next, according to Lemma 6,

$$\left|\mathbb{E}[\overline{X}_n(p)] - \mu_{\star,n}\right| \leqslant \frac{R}{n} \sum_{a=1,a\neq a_*}^{K} \left\{\frac{(1+\varepsilon_0)\log n}{\mathcal{K}^{(N)}(F_a, \mu_{\star})} + o(\log n) + O(1)\right\} = O(n^{-1}), \qquad (69)$$

that leads to

$$\mathbb{P}\big(\big|\overline{X}_n(p) - \mu_{\star,n}\big| \geqslant \varepsilon/2\big) \leqslant \frac{\big|\mathbb{E}[\overline{X}_n(p)] - \mu_{\star,n}\big|}{\varepsilon/2} = \frac{O(n^{-1})}{\varepsilon/2}. \tag{70}$$

Therefore, $\exists c > 0, \forall 0 < \varepsilon < 1$ such that

$$\mathbb{P}(\big|\overline{X}_n(p) - \mu_{\star,n}\big| \geqslant \varepsilon/2) \leqslant cn^{-1}\varepsilon^{-2}, \tag{71}$$

which means

$$\forall n \geqslant N_0, \forall 0 < \varepsilon < 1, \exists c > 0 \text{ that } \mathbb{P}\left(|\overline{X}_n(p) - \mu_{\star}| > \varepsilon\right) \leqslant cn^{-1}\varepsilon^{-2}.$$

Now we see that $|\overline{X}_n(p) - \mu_{\star}| \leqslant R$. With $\varepsilon > \max\{1, R\}$, we have $|\overline{X}_n(p) - \mu_{\star}| > \varepsilon \Leftrightarrow |\overline{X}_n(p) - \mu_{\star}| > R$, therefore the inequality holds as

$$\mathbb{P}\left(|\overline{X}_n(p) - \mu_{\star}| > \varepsilon\right) = 0 \leqslant cn^{-1}\varepsilon^{-2}.$$

with $0 < \varepsilon < \max\{1, R\}, 1 \leqslant n < N_0 \Rightarrow n\varepsilon < \max\{1, R\}N_0 \Rightarrow n^{-1}\varepsilon^{-1} > 1/\max\{1, R\}N_0$. Therefore

$$\forall C > 1/\max\{1, R\}N_0 \Rightarrow \mathbb{P}\left(|\overline{X}_n(p) - \mu_{\star}| > \varepsilon\right) \leqslant 1 < Cn^{-1}\varepsilon^{-1} < Cn^{-1}\varepsilon^{-2},$$

which means

$$\forall n \geqslant 1, \forall 0 < \varepsilon < 1, \exists C > 0 \text{ that } \mathbb{P}\left(|\overline{X}_n(p) - \mu_{\star}| > \varepsilon\right) \leqslant Cn^{-1}\varepsilon^{-2}.$$

That concludes the proof. $\qquad\square$

## F. Convergence of Wasserstein Monte-Carlo tree search

We start with Lemma 8, which shows the concentration of empirical Q value at any internal node in the tree. This plays an important role in the analysis of our MCTS algorithm.

From the results of Lemma 8 and Theorem 2, we derive Propostion 3 which shows the concentration of any internal V-node and Q-node in the tree. Finally, we get the expected simple bias with convergence rate of $\mathcal{O}(n^{-1/2})$ in Theorem 1.

Let us start with Lemma 8.

**Lemma 8.** *(Lemma 1(Dam et al., 2024b)) For $m \in [M]$, let $(\widehat{V}(m, n))_{n \geqslant 1}$ be a sequence of estimator satisfying*

$$\mathbf{Pr}\left(\left|\widehat{V}(m, n) - V(m)\right| > \varepsilon\right) \leqslant Cn^{-1}\varepsilon^{-2}$$

*Assume that there exists a constant $L > 0$ such that $L = \text{supremum}\{\widehat{V}(m, n)\}_{n \geqslant 1}$. Let $R_i$ be an iid sequence with mean $\mu$ and $S_i$ be an iid sequence from a distribution $p = (p_1, \ldots, p_M)$ supported on $\{1, \ldots, M\}$. Introducing the random variables $N_m^n = \#|\{i \leqslant n : S_i = s_m\}|$, we define the sequence of estimator*

$$\overline{Q}(n) = \frac{1}{n}\sum_{i=1}^{n} R_i + \gamma \sum_{m=1}^{M} \frac{N_m^n}{n} \widehat{V}(m, N_m^n).$$

*Then there exists some constant $c'$ (which depends on $p_i$ (i=1,2,...,M), $\gamma$, $\mu$) such that*

$$\mathbf{Pr}\left(\left|\overline{Q}(n) - \mu - \sum_{m=1}^{M} p_m V(m)\right| \geqslant \varepsilon\right) \leqslant Cn^{-1}\varepsilon^{-2}.$$

Based on the results of the described nonstationary multi-armed bandit problem, we derive theoretical results for `W-MCTS`.

We derive Proposition 3, which shows the polynomial concentration of the estimated mean of the Q-value function at the root node. In Proposition 3, we also show that the estimated mean of the V-value function at the root node converges polynomially to the optimal mean. Based on Proposition 3, we derive the result in Theorem 1, which shows the bias of the expected payoff of the power mean backup at the root node.

At any node of state $s$ at depth $h$ in the tree, the mean of the Q value function, and the mean value of the optimal value function are defined as

$$\widetilde{Q}(s_h, a) = R(s_h, a) + \gamma \widetilde{V}(s_{h+1}), \tag{72}$$

$$\widetilde{V}(s_h) = \underset{a}{\operatorname{argmax}} \widetilde{Q}(s_h, a), \tag{73}$$

with $h = [H-1, ..., 1, 0]$, $\widetilde{V}(s_h)$ is the value return from rollouts at state $s_h$, $R(s_h, a)$ is the mean reward received at state $s_h$ after taking action $a$. Let us denote $a_{k^*}$ as the optimal action at the root node.

**Proposition 3.** *When we apply the* `W-MCTS` *algorithm to an MCTS tree of depth $(H)$, at any depth $h$ of the tree, we have*

*(i) At any depth $h$, $\exists$ constant $C_0 > 0$ that for any $0 < \varepsilon < 1, n \geqslant 1$, we can derive*

$$\mathbf{Pr}\left( \left| \overline{V}_m(s_h, a_k, n) - \widetilde{V}(s_h, a_k) \right| \geqslant \varepsilon \right) \leqslant C_0 n^{-1} \varepsilon^{-2}. \tag{74}$$

*(ii) At any depth $h$, $\exists$ constant $C_0 > 0$ that for any $0 < \varepsilon < 1, n \geqslant 1$, we can derive*

$$\mathbf{Pr}\left( \left| \overline{Q}_m(s_h, a_k, n) - \widetilde{Q}(s_h, a_k) \right| \geqslant \varepsilon \right) \leqslant C_0 n^{-1} \varepsilon^{-2}. \tag{75}$$

*Proof.* We will prove this by induction on the depth $D$ of the tree.

**Base case (depth $H = 1$):**
At depth 1, the tree consists of only the root node. The state at the root is denoted by $s_0$. At time step $t$, suppose the agent takes action $a_k$ at $s_0$, resulting in an intermediate reward $r_t(s_0, a_k)$, and transitions to the next state $s_1$.

We assume that the reward $R(s_0, a_k)$ represents the mean reward received at state $s_0$ after taking action $a_k$.

We recall the definition of $\widetilde{Q}(s_0, a_k)$, defined as

$$\widetilde{Q}(s, a) = R(s, a) + \gamma \widetilde{V}(s). \tag{76}$$

where $V_m(s_1)$ is the value of the rollout policy at state $s_1$, $\mathcal{A}_{s_0}$ is the set of feasible actions at state $s_0$, $|\mathcal{A}_{s_0}| = M$, $\mathbb{P}(s_1|s_0, a_k)$ is the probability transition of taking action $a_k$ at state $s_0$ to state $s_1$. We have

$$\overline{Q}_m(s_0, a_k, n) = \frac{1}{n} \sum_{t=1}^{n} r_t(s_0, a_k) + \gamma \sum_{s_1 \sim \tau(s_0, a_k)} \frac{T_{s_0, a_k}^{s_1}(n)}{n} \overline{V}_m(s_1, T_{s_0, a_k}^{s_1}(n))$$

Equation (74) is a direct result of Lemma 8, where $X_t$ represents the intermediate reward $r_t(s_0, a_k)$ at time step $t$. The probability distribution $p = (p_1, p_2, \ldots, p_M) \sim \mathbb{P}(\cdot|s_0, a_k)$, where $\mathbb{P}(\cdot|s_0, a_k)$ is the transition probability dynamic for taking action $a_k$ in state $s_0$.

For each $m \in [M]$, the sequence $(\overline{V}_{m,t})_{t \geqslant 1}$ at time step $t$ corresponds to the deterministic initial value function $\widetilde{V}_m(s_1)$, where:

$$\mathbf{Pr}\left( \left| \overline{V}_m(s_m, n) - \widetilde{V}(s_1) \right| > \varepsilon \right) \leqslant C n^{-1} \varepsilon^{-2},$$

with $m = 1, 2, 3, \ldots, M$, and $s_m \sim \tau(\cdot|s_0, a_k)$. Here, $\tau(\cdot|s_0, a_k)$ denotes the transition kernel from state $s_0$ to $s_m$, given action $a_k$.

Equation (75) is the direct results from Theorem 2. In detail, we have from equation (74),

$$\mathbf{Pr}\left( \left| \overline{Q}_m(s_0, a_k, n) - \widetilde{Q}(s_0, a_k) \right| > \varepsilon \right) \leqslant C n^{-1} \varepsilon^{-2}, \text{ with } a_k \in \mathcal{A}_{s_0}$$

Because by definition:

$$\widetilde{V}(s_0) = \max_{a_k \in \mathcal{A}_{s_0}} \widetilde{Q}(s_0, a_k) \tag{77}$$

$$\overline{V}_m(s_0, n) = \left( \sum_{a \in \mathcal{A}_{s_0}} \frac{T_{s_0,a}(n)}{n} \left( \overline{Q}_m(s_0, a, T_{s_0,a}(n)) \right)^p \right)^{\frac{1}{p}} \quad \text{with } p \in [1, +\infty) \tag{78}$$

Then we have

$$\mathbf{Pr} \left( \left| \overline{V}_m(s_0, n) - \widetilde{V}(s_0) \right| > \varepsilon \right) \leqslant Cn^{-1}\varepsilon^{-2}$$

that concludes for Equation (75)

**Let us assume that for a tree of depth** $H - 1$, the theorem holds for all its children.

Now, consider a tree with depth $H$. When an action is taken at the root node, where the state is $s_0$, the tree transitions into a subtree of depth $H$. By the induction hypothesis, the results hold for any internal node of the tree after taking the first action.

We have $s_1 \sim \tau(s_0, a_k)$, where $\tau(s_0, a_k)$ denotes the transition dynamics. By definition, the value function at the leaf nodes is $\widetilde{V}(s_H) = V_0(s_H)$, and for all $h \leqslant H - 1$, the following holds:

$$\widetilde{Q}(s_h, a) = R(s_h, a) + \gamma \sum_{s_{h+1} \in \mathcal{A}_s} \mathbb{P}(s_{h+1} \mid s_h, a)\widetilde{V}(s_{h+1}),$$

$$\widetilde{V}(s_h) = \max_a \widetilde{Q}(s_h, a),$$

where $R(s_h, a)$ represents the immediate reward at state $s_h$ after taking action $a$, $\gamma$ is the discount factor, and $\mathbb{P}(s_{h+1} \mid s_h, a)$ is the probability of transitioning to state $s_{h+1}$ from $s_h$ by taking action $a$.

By the assumption of the induction the root node of a subtree with depth $(H - 1)$ at state $s_1$ we have

$$\mathbf{Pr} \left( \left| \overline{V}_m(s_1, n) - \widetilde{V}(s_1) \right| > \varepsilon \right) \leqslant Cn^{-1}\varepsilon^{-2}$$

(75) Let's apply Lemma 8 with $\{X_t\}$ is the intermediate reward $\{r_t(s_0, a_k)\}$, $p = (p_1, p_2, ...p_M) \sim \mathbb{P}(\cdot|s_0, a_k)$. For $m \in [M]$, each $(\overline{V}_{m,t})_{t \geqslant 1}$ at time step t is the empirical Value function $\overline{V}_t(s_1)$. We will have

$$\mathbf{Pr} \left( \left| \overline{Q}_m(s_0, a_k, n) - \widetilde{Q}(s_0, a_k) \right| > \varepsilon \right) \leqslant Cn^{-1}\varepsilon^{-2}, \text{ with } a_k \in \mathcal{A}_{s_0}$$

(74) follows the results of Theorem 2 as at the root node $s_0$ of depth $H$, with

$$\widetilde{V}(s_0) = \max_{a_k \in \mathcal{A}_{s_0}} \widetilde{Q}^{(0)}(s_0, a_k) \tag{79}$$

$$\overline{V}_m(s_0, n) = \left( \sum_{a \in \mathcal{A}_s} \frac{T_{s_0,a}(n)}{n} \left( \overline{Q}_m(s_0, a, T_{s_0,a}(n)) \right)^p \right)^{\frac{1}{p}} \quad \text{for some } p \in [1, +\infty) \tag{80}$$

And because

$$\mathbf{Pr} \left( \left| \overline{Q}_m(s_0, a_k, n) - \widetilde{Q}(s_0, a_k) \right| > \varepsilon \right) \leqslant Cn^{-1}\varepsilon^{-2}, \text{ with } a_k \in \mathcal{A}_{s_0}$$

Then, we have

$$\mathbf{Pr} \left( \left| \overline{V}_m(s_0, n) - V(s_0) \right| > \varepsilon \right) \leqslant Cn^{-1}\varepsilon^{-2}.$$

that concludes for (74)

The results of Proposition 3 hold for any node in the tree with the tree of depth $(H)$. By induction, we can conclude the proof. □

**Theorem 1.** *We have at the root node $s_0$,*

$$\left|\mathbb{E}[\overline{V}_m(s_0, n)] - V(s_0)\right| \leqslant \mathcal{O}(n^{-1/2}).$$

*Proof.* Using the convexity of $f(x) = |x|$ and applying Jensen's inequality we have

$$\left|\mathbb{E}[\overline{V}_m(s_0, n)] - V(s_0)\right| \leqslant \mathbb{E}[\left|\overline{V}_m(s_0, n)] - V(s_0)\right|]$$

$$= \int_0^{+\infty} \mathbb{P}\left(\left|\overline{V}_m(s_0, n) - V(s_0)\right| \geqslant s\right) ds$$

$$\leqslant \int_0^{n^{-\frac{1}{2}}} 1 ds + \int_{n^{-\frac{1}{2}}}^{+\infty} C_0 n^{-1} s^{-2} ds$$

$$\leqslant n^{-\frac{1}{2}} + C_0 n^{-1} \left(\frac{s^{-2+1}}{-2+1}\right)\Big|_{n^{-\frac{1}{2}}}^{+\infty}$$

$$= (\frac{C_0}{2-1} + 1)n^{-\frac{1}{2}}.$$

Then,

$$\left|\mathbb{E}[\overline{V}_m(s_0, n)] - V(s_0)\right| \leqslant \mathcal{O}(n^{-1/2}).$$

That concludes the proof.

$\square$

