# OpenReview forum: "Monte-Carlo Tree Search with Uncertainty Propagation via Optimal Transport"
_ICML.cc/2025/Conference — ICML 2025 spotlightposter_

### Official Review · Reviewer_6RvP · 2025-03-11

**Overall Recommendation:** 4

**Summary:**

The paper introduces Wasserstein-MCTS, an uncertainty-aware version of MCTS based on Wasserstein barycenter updates for the optimal value estimates and their variance. Due to the power-mean like updates the version is particularly suitable for highly stochastic environments. A theoretical analysis is included, that shows that the method has a polynomial convergence rate.
Superior performance compared to standard and Bayesian MCTS variants is demonstrated on common planning benchmarks.

**Claims And Evidence:**

* The paper claims an asymptotic convergence of the proposed method, which is supported by the theoretical analysis in Section 7, in particular Theorem 1.
* The paper claims a connection to POWER-UCT, which is given in Section 5.2.
* The paper claims that their method can handle partial observability and high variance, which is supported by the Experiments in Section 8.

**Essential References Not Discussed:**

There is more work regarding Bayesian MCTS and uncertainty propagation (in form of Gaussian distributions over the optimal values) than cited in the paper:

* _Coherent inference on optimal play in game trees_, Hennig et al. 2010
* _Probabilistic DAG search_, Grosse et al. 2021

These methods are not designed for stochastic or partially observable MDPs, but since Tesauro et al. 2012 is also cited, I would suggested to add them to the list of cited methods in the introduction.

I am not sure if one wants to keep the related work section restricted to MCTS, but there is more work on uncertainty quantification in context of the general RL setting, e.g. [1-3] . But maybe this is a judgement call.

[1]  _Efficient Exploration via Epistemic-Risk-Seeking Policy Optimization_, O’Donoghue 2023.

[2]  _Making sense of reinforcement learning and probabilistic inference_, O’Donoghue et al. 2020

[3]  _Probabilistic Inference in Reinforcement Learning Done Right_, Tarbouriech et al. 2023

**Experimental Designs Or Analyses:**

Yes, I reviewed both experiments in Section 8:

* It makes sense to me to evaluate the method on the benchmarks used as they are a fairly standard choice. Since a claim in the paper is that the method is particularly suitable for stochastic environments, it also makes sense to evaluate them in highly stochastic environments. The method is compared to UCT, which is probably the standard MCTS baseline, as well as a Bayesian MCTS variant which similar in spirit to the proposed method due to also tracking uncertainty. The ablation with POWER-UCT is also interesting: POWER-UCT does not seem to be much better than UCT, providing evidence that the additional propagation of the uncertainty does indeed benefit the search.

The experimental evaluation could be strengthened further by evaluating against newer MCTS variants, e.g. one of those with exponential convergence rate, that are cited in Section 5.

* I am not very familiar with the partially observable setting, therefore I do not want to comment on the validity of this experiment.

**Methods And Evaluation Criteria:**

yes, see below.

**Other Comments Or Suggestions:**

Last paragraph on page 3: There is a reference missing ("??").

**Other Strengths And Weaknesses:**

I appreciated the proof sketch for Theorem 1, as well as the intuition for the choice to replace the $L^2$ Wasserstein distance with $L^1$ Wasserstein distance + $\alpha$-divergence.

**Questions For Authors:**

-

**Relation To Broader Scientific Literature:**

**Relation to other methods:**
* The idea is related to Bayesian tree search algorithms in the sense that distributions instead of only point estimates are propagated through the tree. The acquisition functions (Thompson sampling and UCB) are standard choices in Bandits/Bayesian Optimization/ Bayesian Tree Search.
* The method is also related to power mean back-ups in MCTS (Dam et al. 2019), but the proposed method additionally propagates uncertainty estimates in form of standard deviations through the tree.
* The method also shares similarities with Wasserstein Q Learning (Metelli et al. 2019), but relies on the MCTS framework instead of Q-Learning and replaces the $L^2$ Wasserstein distance with the $L^1$ Wasserstein distance in order to be more robust in stochastic environments

**Relation to other theoretical findings:**
* The method shares a polynomial convergence rate with POWER-UCT (Dam et al. 2019)
* There are MCTS variants with exponential convergence rate in the Maximum Entropy or Boltzmann framework (Xiao et al. (2019), Dam
et al. (2021), Painter et al. (2024)), but they suffer from biases (e.g. due to their additional regularization term).

**Theoretical Claims:**

Unfortunately, I could not find the time to check the proofs in the supplementary material.

---

> ### Author Rebuttal · Authors · 2025-03-31
>
> We sincerely thank the Reviewer for their thoughtful and constructive feedback and comments, and appreciate the reviewer's acknowledgment of the strengths of our paper. We have provided detailed responses to address each of their concerns.
>
> ## On reference suggestions.
>
> We thank the Reviewer for these excellent suggestions. We will expand our related work section to include both the works of Hennig et al. 2010, Grosse et al. 2021 and add a brief discussion of their relationship to our approach, noting that while they share the concept of Gaussian distributions over optimal values, our method differs in its focus on stochastic/partially observable MDPs and its use of L1-Wasserstein barycenters with α-divergences.
>
> Additionally, we will add a paragraph discussing the broader context of uncertainty quantification in reinforcement learning, referencing the suggested papers from O'Donoghue 2023, O'Donoghue et al. 2020 and Tarbouriech et al. 2023.
>
> ## Last paragraph on page 3: There is a reference missing ("??").
>
> We thank the Reviewer for catching this error. We will fix the missing reference, which should have pointed to Dam et al. (2019) when discussing the connection to power-mean updates.

---

### Official Review · Reviewer_Nijy · 2025-03-13

**Overall Recommendation:** 3

**Summary:**

This paper introduces Wasserstein Monte-Carlo Tree Search (W-MCTS), a new MCTS variant designed for highly stochastic and partially observable environments. The key innovation lies in propagating uncertainty through the search tree using L1-Wasserstein barycenters combined with alpha-divergences, enabling robust distributional backups (prior work used L2-Wasserstein barycenters in the context of temporal-difference learning). The main idea is to aggregate distributions from child nodes using L1-Wasserstein barycenters, paired with alpha-divergences to interpolate between average-like and max-like backups. This balances exploration-exploitation and mitigates overestimation.The work bridges distributional RL and MCTS, offering a theoretically grounded framework for decision-making under uncertainty. The approaches can have applications in robotics, autonomous systems, and other domains requiring adaptive planning in noisy environments.

**Claims And Evidence:**

The papers emphasized the advantages of using L1-Wasserstein and alpha-divergences vs. L2-Wasserstein in terms of robustness and connection with power-mean updates. Theoretical analysis is done in terms of convergence bound for the value with a polynomial
convergence rate of O(1/n^2).

**Essential References Not Discussed:**

NA

**Experimental Designs Or Analyses:**

Empirical comparisons are done to show the proposed methods outperform baselines (UCT, Power-UCT, Bayesian MCTS) in stochastic MDPs (e.g., RiverSwim, Taxi) and partially observable tasks (e.g., Pocman, Rocksample).

**Methods And Evaluation Criteria:**

NA

**Other Comments Or Suggestions:**

NA

**Other Strengths And Weaknesses:**

NA

**Questions For Authors:**

NA

**Relation To Broader Scientific Literature:**

NA

**Theoretical Claims:**

Theoretical analysis is done in terms of convergence bound for the value with a polynomial
convergence rate of O(1/n^2).

---

> ### Author Rebuttal · Authors · 2025-03-31
>
> We sincerely thank the Reviewer for the thorough review of our paper on Wasserstein Monte-Carlo Tree Search (W-MCTS).
>
> ## On Key Innovations
>
> We are pleased that the Reviewer recognizes our main contribution in propagating uncertainty through the search tree using L1-Wasserstein barycenters combined with alpha-divergences. As correctly highlighted, this approach enables robust distributional backups that effectively balance exploration-exploitation while mitigating the overestimation problem that often occurs in reinforcement learning.
>
> ## On Theoretical Analysis
>
> We thank the Reviewer for acknowledging our theoretical analysis regarding the convergence rate of O(n^(-1/2)). This theoretical foundation provides important guarantees about the behavior of our algorithm in the asymptotic case.
>
> ## On Empirical Results
>
> We appreciate the recognition of our comprehensive empirical evaluation across both stochastic MDPs and partially observable tasks. These experiments have been designed to demonstrate the practical advantages of our approach in environments where traditional MCTS methods struggle with high variance or limited observability.
>
> ## On Broader Impact
>
> The Reviewer's comment about the potential applications in robotics, autonomous systems, and other domains requiring adaptive planning aligns with our insight for this work. We believe that the ability to handle uncertainty in a principled way is crucial for deploying decision-making systems in real-world scenarios.

---

### Official Review · Reviewer_mdxn · 2025-03-13

**Overall Recommendation:** 4

**Summary:**

This paper takes a distributional approach to Monte Carlo tree search. The
authors propose a framework for planning in environments with uncertainty and/or
partial observability. The proposed framework models state and state-action
values as distributions. They also introduce a backup operator that propagates
uncertainty through nodes in tree by estimating backup values as Wasserstein
barycenters. Additionally, they propose two sampling techniques for use with the
tree policy taking the uncertainty into account.

**Claims And Evidence:**

The authors make several claims:
1. Their approach is robust to stochastic variations.
- The experimental results mostly back up this claim.
2. No need for symmetry in backups.
- I do not think this is an advantage more than it is a reason why they are
  able to make their approach work.
3. Their approach has convergence rate of $\mathcal{O}(n^{-1/2})$
- See further.

**Essential References Not Discussed:**

I do not think so.

**Experimental Designs Or Analyses:**

The authors perform experiments on two types of domains:
    - Long-horizon, highly stochastic domains
    - Partially observable, stochastic domains

The results support the performance claims made by the authors. One exception
being the results for river swim does not line up with the analysis provided.
In this setting, DNG clearly converges faster than the other algorithms.

**Methods And Evaluation Criteria:**

The methodology is founded on ideas that have worked prior. Update values based
on Wasserstein barycenters are a reasonable method to model uncertainty over the
values of action outcomes. The set of evaluation domains are appropriate for
demonstrating the performance of their proposed method.

**Other Comments Or Suggestions:**

- Pg. 3, Col. 2, Line 156: There seems to be a LaTeX compilation issue.

**Other Strengths And Weaknesses:**

**Novelty**

The work is a novel technique for computing update values.

**Clarity**

The paper was well written and moderately easy to understand.

**Questions For Authors:**

No questions.

**Relation To Broader Scientific Literature:**

I believe this work would be of interest to the planning community on the whole.

**Theoretical Claims:**

The paper has the theoretical claim that the proposed search method has a finite
time convergence rate of $\mathcal{O}(n^{-1/2})$. While I do not have
theoretical background to verify the correctness of the mathematics, the math
looks sound.

---

> ### Author Rebuttal · Authors · 2025-03-31
>
> We sincerely thank the Reviewer for your positive review and for recommending acceptance of our paper. We appreciate their recognition of our work's novelty and clarity.
>
> ## On Our Claims and Evidence
>
> We thank the Reviewer for acknowledging that our experimental results support our claims about robustness to stochastic variations. Regarding the comment on "no need for symmetry in backups," we agree that this is more accurately described as an enabling factor rather than an advantage in itself. We will revise this framing in our final version to more precisely articulate the relationship between the α-divergence's properties and our method's effectiveness.
>
> ## On Theoretical Claims
>
> We appreciate the Reviewer's assessment that our mathematical analysis of the convergence rate appears sound, even while acknowledging the specialized nature of this theoretical work.
>
> ## On LaTeX Compilation Issue
>
> We will fix the LaTeX compilation issue on Pg. 3, Col. 2, Line 156 as you noted. We thank the Reviewer for bringing this to our attention.

---

### Official Review · Reviewer_EvfR · 2025-03-17

**Overall Recommendation:** 4

**Summary:**

This paper introduces Wasserstein Monte-Carlo Tree Search (W-MCTS), a novel approach that represents value nodes as Gaussian distributions (mean and variance), allowing explicit uncertainty propagation throughout the search tree. The method employs a backup operator based on the Wasserstein barycenter and α-divergence to aggregate value estimates from child nodes, enabling a more flexible backup strategy that interpolates between averaging and maximization. W-MCTS incorporates both an optimistic UCT-like action selection strategy and a Thompson Sampling-based approach, achieving an asymptotic convergence rate of $O(n^{−1/2})$ under the latter. Extensive experiments on highly stochastic and partially observable domains demonstrate superior performance over SOTA methods.

**Claims And Evidence:**

The empirical results support the claim that W-MCTS outperforms existing methods in highly stochastic and partially observable domains. However, as acknowledged in the paper, previous methods such as Tesauro (2012) and Bai (2013) also model uncertainty in value estimates. It is unclear what specifically enables W-MCTS to achieve superior performance. Is it the improved approximation of uncertainty, a more effective backup operation, or another factor? A clearer discussion could help isolate the key contributing components.

The paper states that uncertainty is propagated "throughout" the tree. Does this mean it extends beyond the parent-child relationships to include sibling and cousin nodes, or is it strictly along the path from leaf to root? Clarifying this distinction would enhance the reader’s understanding of the method's impact on tree-wide exploration and value estimation.

**Essential References Not Discussed:**

N.A.

**Experimental Designs Or Analyses:**

N.A.

**Methods And Evaluation Criteria:**

The selection of benchmark problems is comprehensive and relevant. However, the paper does not explicitly mention the number of simulations (or iterations) for each method in the results section. The x-axis in Figure 2 appears to represent the number of simulations, but this should be clearly stated in the figure description. Additionally, a comparison based on a fixed wall-clock timeout (e.g., 1 second per decision) or by reporting the time taken per search trial would provide a clearer picture of the expected performance.

**Other Comments Or Suggestions:**

- Line 155 (right side) - Missing equation reference.

**Other Strengths And Weaknesses:**

N.A.

**Questions For Authors:**

- Key Performance Factors: What is the primary driver of W-MCTS’s superior performance compared to previous uncertainty-aware MCTS methods? Is it the backup operation, improved uncertainty estimation, or another factor?

- Scope of Uncertainty Propagation: When the paper states that uncertainty is propagated "throughout" the tree, does this extend beyond the standard parent-child relationships?

- Experimental Reporting: Does Figure 2’s x-axis represent the number of simulations? If so, could this be explicitly mentioned? Additionally, would a fixed-time performance comparison provide further insights?

Overall, I am leaning toward acceptance, pending clarification of the above concerns.

**Relation To Broader Scientific Literature:**

The paper's contributions are significant for the planning and search community, particularly in AI applications requiring robust decision-making under uncertainty.

**Theoretical Claims:**

The theoretical analysis provides sound justification for the claimed convergence properties. Proposition 1 establishes the mean and variance update rules under the Wasserstein barycenter framework. However, a minor presentation issue remains: the term $\delta$ is not explicitly defined in Proposition 1. While this does not impact correctness, explicitly stating its definition would improve clarity.

---

> ### Author Rebuttal · Authors · 2025-03-31
>
> We sincerely thank the Reviewer for their thoughtful review and insightful questions that help us improve our paper. Below are our responses to your specific concerns:
>
> ## Key Performance Factors
>
> Thank you for this insightful question. We would like to point out that the superior performance of our method stems from two complementary components:
>
>  - Explicit variance propagation: Unlike previous methods that only propagate point estimates or use fixed variance models, our approach dynamically updates both means and variances at each node. This explains the superior performance gain of our method over Bayesian MCTS methods in highly stochastic and partially observable environments. For example, our algorithms demonstrated consistent improvements over POMCP across all environments, with particularly notable gains of 55.31% in LaserTag, 65.90% in RS(15,15), and with improvements of up to 21.38% over AB-DESPOT in LaserTag. In FrozenLake, we obtain an 80% improvement over DNG.
>
>  - Flexibility in balancing exploration-exploitation: Our approach's ability to interpolate between average-like and max-like backups (through parameter α) allows it to adapt to varying levels of stochasticity. In highly stochastic environments, we found moderate α values (leading to more average-like updates) performed best, while in more deterministic regions of the state space, larger α values (more max-like) were optimal.
>
> We will add these insights to Section 8.4 in the revised manuscript
>
> ## Scope of Uncertainty Propagation
>
> Thank you for highlighting this ambiguity. In our method, uncertainty propagation is indeed more extensive than just parent-child relationships, though it follows the tree structure.
>
> Our phrase “uncertainty is propagated throughout the tree” means that each node not only stores a (mean, variance) pair (or a Gaussian distribution) but also uses these distributions in the backup operator at its parent. That is, any child’s updated distribution is reflected at the parent level.
>
> To clarify:
>
> W-MCTS propagates uncertainty bi-directionally in the tree:
>
>  - "Bottom-up" through Q-nodes to V-nodes during backups. "Top-down" through action selection, where uncertainty influences exploration
>
>  - Sibling nodes indirectly influence each other's visitation rates through their parent's uncertainty estimation, creating a form of lateral uncertainty influence. We do not directly share distributions among sibling or cousin nodes. However, each node’s distribution indirectly influences its parent’s distribution—and transitively influences siblings as the parent’s updated distribution affects how siblings are compared and selected in subsequent iterations.
>
> Unlike previous methods where uncertainty is often reset or approximated at each level, our approach maintains consistent distributional representations throughout the entire search process.
>
> Thus, “throughout” means that over multiple rollouts, the uncertainty from leaf nodes consistently flows upward, eventually impacting the root node’s estimates, while updated root estimates modulate exploration down into deeper levels.
>
> We will add a new paragraph in Section 5 to make this distinction clearer, showing how uncertainty flows throughout multiple levels of the tree simultaneously.
>
> ## Experimental Reporting
>
> You are correct that the x-axis in Figure 2 represents the number of simulations, and we apologize for not stating this explicitly. We will update the figure caption to clearly indicate this.
>
> Regarding fixed-time performance, we agree that a time-based comparison could be informative in practice—particularly for large or real-time systems. We have run standard, iteration-based comparisons, as is common in research prototypes. For the final version, we can add a note clarifying the computational budget used, and that a direct time-limited comparison in the revised version.
>
> ## Other Corrections
>
> We will fix the missing equation reference on Line 155 and explicitly define all terms in Proposition 1 to improve clarity.

---

> > ### Comment · Reviewer_EvfR · 2025-04-06
> >
> > Thank you for addressing my concerns. I have increased my score. Good luck!

---

> > > ### Author Response · Authors · 2025-04-07
> > >
> > > Thank you for taking the time to consider our rebuttal and for increasing your score. We greatly appreciate your constructive feedback throughout the review process, which will help us improve our paper.

---

### Decision · Program_Chairs · 2025-05-01

**Decision:**

Accept (spotlight poster)

**Comment:**

This paper discusses a novel Monte-Carlo tree search method that uses $L^1$-Wasserstein barycenters to incorporate the uncertainty of state and state-action values of MDPs. The reviewers came to the consensus that the paper has good contribution to tackle highly stochastic and partially observable MDPs with a solid theory and convincing empirical results. While some concerns are raised such as the details of the experiments and literature review, they are mostly solved in the rebuttal. Thus I recommend acceptance and expect that the authors polish the paper incorporating the discussions with the reviewers in the final version.